# Multiscale, multimodal analysis of tumor heterogeneity in IDH1 mutant vs wild-type diffuse gliomas

**Michael E. Berens**[1☯*], **Anup Sood**[2☯*], **Jill S. Barnholtz-Sloan**[3☯], **John F. Graf**[2], **Sanghee Cho**[2], **Seungchan Kim**[4], **Jeffrey Kiefer**[1¤a], **Sara A. Byron**[1], **Rebecca F. Halperin**[1], **Sara Nasser**[1], **Jonathan Adkins**[1], **Lori Cuyugan**[1], **Karen Devine**[3], **Quinn Ostrom**[3¤b], **Marta Couce**[3], **Leo Wolansky**[3¤c], **Elizabeth McDonough**[2], **Shannon Schyberg**[2], **Sean Dinn**[2], **Andrew E. Sloan**[5], **Michael Prados**[6], **Joanna J. Phillips**[6], **Sarah J. Nelson**[7†], **Winnie S. Liang**[1], **Yousef Al-Kofahi**[2], **Mirabela Rusu**[2¤d], **Maria I. Zavodszky**[2¤e], **Fiona Ginty**[2*]

**1** Translational Genomics Research Institute, Phoenix, AZ, United States of America, **2** GE Research Center, Niskayuna, NY, United States of America, **3** Department of Population and Quantitative Health Sciences and Case Comprehensive Cancer Center, Case Western Reserve University School of Medicine, Cleveland, OH, United States of America, **4** Department of Electrical and Computer Engineering, Roy G. Perry College of Engineering, Prairie View A&M University, Prairie View, TX, United States of America, **5** Department of Neurosurgery, University Hospitals-Seidman Cancer Center, Cleveland, OH, United States of America, **6** Department of Neurological Surgery, Helen Diller Cancer Center, University of California San Francisco, San Francisco, CA, United States of America, **7** Department of Radiology and Biomedical Imaging, University of California, San Francisco, CA, United States of America

☯ These authors contributed equally to this work.
† Deceased.
¤a Current address: Systems Oncology, Scottsdale, AZ, United States of America
¤b Current address: Department of Medicine, Epidemiology and Population Sciences, Dan L Duncan Comprehensive Cancer Center, Baylor College of Medicine, Houston, TX, United States of America
¤c Current address: Department of Diagnostic Imaging and Therapeutics, UConn Health, Farmington, CT, United States of America
¤d Current address: Stanford University, Department of Radiology, Stanford, CA, United States of America
¤e Current address: Biogen, Cambridge, MA, United States of America
* mberens@tgen.org (MEB); Anup.Sood@ge.com (AS); ginty@research.ge.com (FG)

**Data Availability Statement:** Data underlying the study are available on dbGaP: accession number phs001460.v1.p1.

## Abstract

Glioma is recognized to be a highly heterogeneous CNS malignancy, whose diverse cellular composition and cellular interactions have not been well characterized. To gain new clinical- and biological-insights into the genetically-bifurcated IDH1 mutant (mt) vs wildtype (wt) forms of glioma, we integrated data from protein, genomic and MR imaging from 20 treatment-naïve glioma cases and 16 recurrent GBM cases. Multiplexed immunofluorescence (MxIF) was used to generate single cell data for 43 protein markers representing all cancer hallmarks, Genomic sequencing (exome and RNA (normal and tumor) and magnetic resonance imaging (MRI) quantitative features (protocols were T1-post, FLAIR and ADC) from whole tumor, peritumoral edema and enhancing core vs equivalent normal region were also collected from patients. Based on MxIF analysis, 85,767 cells (glioma cases) and 56,304 cells (GBM cases) were used to generate cell-level data for 24 biomarkers. K-means clustering was used to generate 7 distinct groups of cells with divergent biomarker profiles and deconvolution was used to assign RNA data into three classes. Spatial and molecular heterogeneity metrics were generated for the cell data. All features were compared between

**Funding:** Funding support for the project was provided by GEGR. Additional support was provided by The Ben & Catherine Ivy Foundation (MEB, JK, SAB, RH, SN, JA, LC, MP and SJN); by the Case Comprehensive Cancer Center Support Grant (P30 CA043703; JS B-S, KD, QO, MC, LW), the Sally S Morley Designated Professorship in Brain Tumor Research (JS B-S) and the Peter Cristal Endowed Professorship in Neurooncological Surgery (AES)); by Texas A&M University Systems' Chancellor's Research Initiative (CRI) Award for the Center for Computational Systems Biology at the Prairie View A&M University (SK). AES is supported by NIH CA217956 and CA236215, as well as the Peter D Cristal Chair, the Center of Excellence for Translational Neuro-Oncology, the Kimble Family Foundation, the Gerald Kaufman Fund for Glioma Research, and the Ferry Family Foundation at University Hospitals of Cleveland.

**Competing interests:** The authors have declared that no competing interests exist.

IDH mt and IDHwt patients and were finally combined to provide a holistic/integrated comparison. Protein expression by hallmark was generally lower in the IDHmt vs wt patients. Molecular and spatial heterogeneity scores for angiogenesis and cell invasion also differed between IDHmt and wt gliomas irrespective of prior treatment and tumor grade; these differences also persisted in the MR imaging features of peritumoral edema and contrast enhancement volumes. A coherent picture of enhanced angiogenesis in IDHwt tumors was derived from multiple platforms (genomic, proteomic and imaging) and scales from individual proteins to cell clusters and heterogeneity, as well as bulk tumor RNA and imaging features. Longer overall survival for IDH1mt glioma patients may reflect mutation-driven alterations in cellular, molecular, and spatial heterogeneity which manifest in discernable radiological manifestations.

## Introduction

Gliomas represent the most common type of malignant brain tumor, comprising 81% of malignant brain and central nervous system (CNS) tumors and 27% of all brain and CNS tumors in the United States[1]. While gliomas are relatively rare in the general population with an average annual age-adjusted incidence of 6.2 per 100,000, these primary brain tumors contribute significant morbidity and mortality, with glioblastoma carrying a 5-year survival rate of less than 6% [1].

The landscape of our knowledge about molecular features required for accurate diagnosis and prognosis for glioma patients has advanced greatly in the last decade [2–5]. Molecular sub-classification highlights different genetic underpinnings of glioblastoma [6], which offer some prognostic insight [7], likely attributable, in part, to gene expression patterns influencing vulnerability to radiation [8]. The World Health Organization (WHO) classifies gliomas into defined categories based upon histologic and molecular features; gliomas are assigned into four grades of increasing aggressiveness. Additionally, the methylation status of O6-methyl-guanine-DNA methyltransferase (MGMT) has been implicated as a useful biomarker for conferring tumor resistance to alkylating chemotherapies; methylation of the MGMT promoter leads to transcriptional silencing of MGMT, which is associated with loss of MGMT expression and increased response to alkylating chemotherapies such as temozolomide (TMZ) [9]. Analysis of DNA methylation from gliomas identified a DNA methylation-based phenotype, G-CIMP, which is characterized by global hypermethylation of CpG islands and is predictive of increased survival; this G-CIMP phenotype is associated with isocitrate dehydrogenase (IDH) mutation status [3, 4, 10].

*IDH* wild type (wt) in histologically defined low-grade gliomas is associated with poor clinical prognosis that more resembles glioblastoma multiforme (GBM), which generally lack *IDH* mutation (IDHmt) [3, 11]. Conversely, *IDH* mutations are observed in the majority of lower-grade gliomas and are associated with better clinical outcomes. In low-grade gliomas with *IDH* mutations, 1p/19q codeletion is further associated with oligodendrogliomas and better chemotherapeutic response [12]. The validation of some of these molecular biomarkers for diagnosis and prognosis has prompted WHO to include molecular subclasses into their latest classification schema for CNS tumors, including addition of MGMT methylation and IDH-mutant/IDH-wildtype classifications for glioblastoma, as well as IDH-mutant and 1p/19q-codeleted classifications for oligodendrogliomas and anaplastic oligodendrogliomas [13].

Intratumoral heterogeneity, even across molecular subtypes, is now also appreciated as a characteristic of glioma and glioblastoma [14] and has been shown to occur temporally [15], spatially [16] [17], for oncogenic drivers [18], and through the stem cell lineage [19]. Heterogeneity features have been identified by radiologic imaging with quantitative features, including distinguishing between IDH1mt vs wt gliomas [20]. While these and other studies have interrogated glioma heterogeneity using bulk transcriptomics and single cell sequencing, medical imaging has also provided valuable heterogeneity insights, albeit limited by resolution (e.g. 1 voxel, the volumetric unit, in a 1.5 T MRI image contains approx. 1–2 million cells). There have been no investigations to date of cell-level spatial heterogeneity in protein expression or cell types and how they relate to the radiological appearance of these tumors on MRI. Understanding malignant progression in IDH1 mt and wt patients at multiple scales and in a spatial context is pivotal to delineating biological events underlying glial tumors and may facilitate tailored treatment approaches as well as reveal new therapeutic targets. Moreover, this multiscale characterization may facilitate the identification of quantitative metrics derived from non-invasive imaging, i.e. MRI, which correlate with or predict molecular and cellular phenotypes. Such metrics may be evaluated for new patients prior to biopsy or surgery and might inform about the presence of certain cellular characteristics that may affect treatment response or outcome.

To discern multimodal differences in relation to IDHmt status, we conducted a multiscale interrogative workflow which combines multiplexed immunofluorescence and single cell spatial analysis of fixed glioma tissue, bulk genomic tumor sequencing, MR imaging quantitative features of the whole tumor and subregions, and patient outcomes. Multiscale datasets were assembled from treatment-naïve cases of grade 2, 3, and 4 astrocytoma/oligodendroglioma (n = 20, referred as treatment-naïve glioma) as well as from recurrent (previously-treated) grade 4 astrocytoma (glioblastoma) (n = 16, referred as recurrent GBM). Tumor tissue punches from diagnostic paraffin blocks were assembled in duplicate (glioma) or triplicate (recurrent GBM) into tissue microarrays for multiplex immunofluorescence staining [21] using 43 markers to identify cell types and functional states corresponding to cancer hallmarks [22]. Exome sequencing data were processed for mutations, copy number aberrations, as well as insertions and deletions. Deconvolution of gene expression data from bulk tumor specimens afforded comparisons of protein levels and transcript levels across cognate specimens. An expert neuroradiologist (LW) outlined on MRI of the treatment naïve glioma, and of recurrent GBM (SJN), while advanced deep learning methods were utilized to delineate necrotic and enhancing cores, as well as peritumororal edema. Morphologic features assessed the volumes of the different regions and their ratios, while simple features, T1 weighted post contrast (T1 Post), Apparent Diffusion Coefficient (ADC), and Fluid Attenuated Inversion Recovery (FLAIR), were extracted from different MRI protocols.

Various MRI-focused studies [23–26] have investigated the ability of imaging features to predict IDH1 mutational status. Studies focused on assessing the tumor volume, contrast enhancement status [27], Visually AcceSAble Rembrandt Images (Vasari) feature set [28, 29], radiomics features [30] or features that were derived via convolutional neural networks [31], among others and used these to train predictive models of IDH1 mutational status. These studies showed great ability to predict IDH1 mutational status with accuracies as high as 89.1% and area under the receiver operator curves (AUC) of 0.95. Other radiogenomic studies have revealed the correlation of IDH1 mutational status with hypoxia induced angiogenesis and identified that the relative cerebral blood volume (rCBV) MRI was able to predict IDH1 mutations status with an 88% accuracy [32]. Unlike the latter studies that predict IDH1 mutational status, we seek to reveal correlations between MRI derived quantitative features, cellular composition and spatial cellular heterogeneity to understand the mechanism of disease progression

in relation to IDH1 mutational status. Such knowledge could enable creation of predictive models on MRI of disease progression or treatment response without the need for an invasive biopsy.

We show lower cell-level protein expression of most hallmark proteins included in this study in IDH1mt vs wt cases consistent with lower aggressiveness of IDHmt tumors. Further, IDH1mt gliomas, irrespective of grade, showed greater spatial heterogeneity but lower molecular heterogeneity of biomarkers associated with angiogenesis (VEGR2, CD31, SMA, S100A4) and invasion (n-cadherin, cofilin, collagen IV, GFAP and vimentin). Similarly, cell classes derived from deconvolution of bulk gene expression data showed the cell class with high expression of most hallmark genes, particularly those belonging to enabling replicative immortality, evading growth suppressors and inducing angiogenesis, were significantly under represented ($<$10%) in the IDHmt tumors. IDH mutation was co-expressed with ATRX mutations and was mutually exclusive of EGFR and PTEN mutations consistent with known tumor biology. Longer overall survival following diagnosis for IDH1mt glioma patients may reflect generalized altered cellular, molecular and spatial heterogeneity, which is also reflected in the MR images as lower enhancement and higher edema.

## Materials and methods

### Patient cohorts

Cohorts of 20 treatment-naïve gliomas (grades 2, 3, and 4 from the Ohio Brain Tumor Study) and 16 post-treatment recurrent glioblastoma (grade 4 from University of California San Francisco [33]) were retrieved based on appropriate patient consent, availability of suitable MR images, FFPE tissue, and specimens suitable for next-generation sequencing (**Table 1 for summary data for the primary glioma and recurrent GBM patient cases and** S1 **and** S2 **Tables for additional details on available sample types and imaging data**). The Ohio Brain Tumor Study was approved by the University Hospitals of Cleveland Institutional Review Board (UH IRB# Case 1307-CC296). All participants provided written informed consent per the IRB approved protocol: "My data and biological samples may be used for brain tumor genetic research studies and/or genetic testing that may take place in the future as part of different research studies". And, "I agree that my anonymized information from this study may be put into a National Institutes of Health publically available database".

### Workflow for multimodal data generation and integration

Using the methods provided below, three parallel analytical interrogations of the treatment-naive glioma and recurrent cases were pursued: multiparametric MRI feature extraction; multiplexed immunofluorescence tissue imaging at single cell level; and RNA and DNA sequencing. **Fig 1** depicts the overall workflow for this multimodal data generation, including multiple analytical approaches to cluster and differentiate clinically variable phenotypes. Given the different clinical characteristics of the two cohorts and the multimodal nature of the data, our analysis was stratified by cohort, yet we aimed at identifying patterns and associations that were consistent across the two cohorts.

### Multiplexed immunofluorescence imaging of disease and cellular biomarkers

Using the original diagnostic FFPE tissue blocks of each case studied, dual (treatment-naïve glioma) or triplicate punches (recurrent GBM) were selected for tissue microarray (TMA) construction and subsequent multiplex immunofluorescence staining and imaging (MxIF). Two

**Table 1. Patient characteristics.**

| Cohort | Treatment naïve primary glioma patients | Recurrent/Refractory GBM patients |
|---|---|---|
| Patient number | 20 | 16 |
| Median (range) age at diagnosis (years) | 57 (26–77) | 51 (29–66) |
| **Gender** | | |
| Male | 12 | 12 |
| Female | 8 | 4 |
| **Ethnicity** | | |
| Caucasian | 18 | 15 |
| Hispanic, Asian, African American | 0,1,1 | 1,0,0 |
| **Histologic grade** | | |
| II | 5 | - |
| III | 7 | - |
| IV | 8 | 16 |
| **IDH1/2 mutation status** | | |
| Mutant (IDH1 R132H) | 8 | 3 |
| Wildtype | 12 | 13 |
| **1p19q codeletion** | | |
| Codeletion | 4 | - |
| Non-codeletion | 16 | - |
| **Median (range) Survival (days)** | | |
| Grade II | 1120 (420–2326) | - |
| Grade III | 487 (370–2964) | - |
| Grade IV (GBM) | 438 (222–541) | 1031 (396–3771) |

replicate TMA slides were used for the treatment-naïve glioma TMAs and 3 replicate TMAs were used for the recurrent GBM TMAs. Control cores were included on all slides for glioma, prostate, melanoma, lung, breast cancer (two per cancer type) to verify antibody performance. Briefly, the Cell DIVE™ platform (GEHC, Issaquah WA) includes a protocol for antigen retrieval, dye-conjugated antibody staining/inactivation (repeated *n* times), and imaging software for up to 60 biomarkers in a single FFPE tissue section [21]. During the imaging workflow, the biomarker images are automatically processed for illumination correction, image registration using the DAPI image (which is acquired in each round of imaging) and tissue autofluorescence (AF) subtraction (AF is imaged on the unstained sample in each of the first 5 rounds and every fifth round thereafter). The processed images are then segmented into individual cells and biomarker fluorescence intensity value is generated for every cell, along with corresponding cell IDs and coordinates. Cell level data can be analyzed in a number of ways including clustering and spatial analysis (**S1A–S1C Fig**). In the current study, TMAs were stained in the above iterative process using 2–3 dye conjugated antibodies in each staining round, for a total of 43 biomarkers. Selected biomarkers were associated with different cancer hallmarks, cell lineage and cell segmentation [22]. Markers of iron metabolism were also included as ferroptosis is an emerging field of study with mechanistic ties to glioma cell resistance to therapy [34–36]. The detailed process for antibody validation (testing, conjugation, antigen sensitivity and staining verification) is described in **S2 Fig** and described in supplementary information of Gerdes et al [21]. Antibody clones, dye conjugate, staining concentration, staining round and hallmark assignments are provided in **S3 Table**.

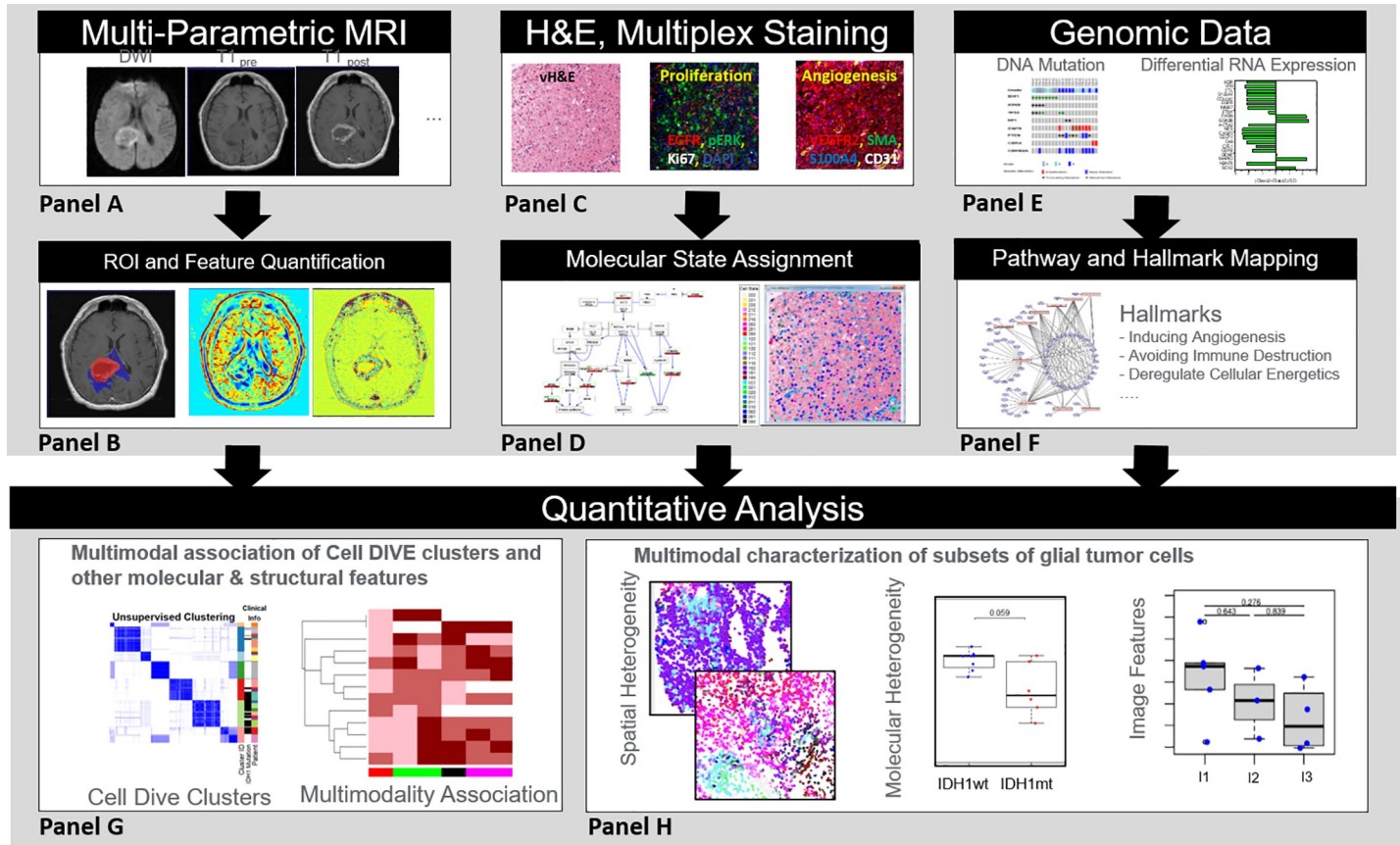

**Fig 1. Overall workflow for generating multiscale, multiparametric data, extraction of various features and/or conversion to higher scales and analysis approaches to differentiate phenotypes.** Multiparametric MRI images (Panels A & B) were segmented for ROIs and various image features to characterize tumor and subregions (necrosis, enhancing and edema) within the tumor. Multiplexed immunofluorescence tissue analysis (MxIF) (Panel C) provided (left-to-right) a virtual H&E (vH&E), which is a pseudo-colored DAPI and AF image, and corresponding overlays of 46 markers (examples shown are for proliferation and angiogenesis markers). Single cell data were generated for every multiplexed marker and intensities were binned into levels for each cell (low, medium or high using the 33$^{rd}$ and 67$^{th}$ quantiles as the thresholds). The molecular "state" of each cell was computed using the ordinal level of each protein. For visualization purposes, the molecular "state" of a cell was overlaid on the vH&E image (Panel D). Genomics data (Panel E & F), including IDH1 mutation status, were summarized into pathways, cancer hallmarks, and enrichments for each tumor. Cell-level biomarker and MRI feature data were clustered across all glioma patients and by IDH 1 status (Panel G) and molecular and spatial heterogeneity were analyzed relative to IDH1 mutation status or tumor grade (Panel H).

## Staining quality assessment and cell segmentation

Staining quality of all multiplexed images was assessed by visual assessment of staining patterns of individual markers in all samples and compared to controls and/or expected patterns. Markers that failed to stain or had non-specific staining or very low or negative expression in both cohorts were excluded from analysis. Since replicate slides were also available, staining intensities were correlated between slides for both treatment-naïve glioma and recurrent GBM cohorts (S3A & S3B Fig). Example stains for two different cores from one patient are shown in (S3C Fig) and show striking consistency.

As described earlier, the single cell analysis workflow consists of cell segmentation and biomarker quantification steps (S1C Fig). First, image background was suppressed using top-hat filtering followed by multi-level image thresholding. Second, nuclei were segmented using a wavelet-based algorithm that uses both nuclei intensity and shape (blobness) information [37]. Nuclear segmentation was followed by whole-cell segmentation, where synthetic cell boundary was extracted by applying Voronoi tessellation using the nuclei as seeds. To avoid producing

very large cells from isolated nuclei, a constraint on the maximum distance between the nucleus and the corresponding cell boundary was applied. Segmented images were visually assessed for segmentation quality and compared with images of DAPI staining and virtual H&E (generated from pseudo-color overlays of DAPI and tissue AF). One image (of 40 total) in the glioma group failed to segment due to poor tissue quality. Five images (of 46 total) from recurrent GBM group were removed from further analysis as the cores contained few (<10%) tumor cells, or were cauterized.

Cell segmentation was followed by quantification of biomarker intensities in each cell. This data, along with cell IDs and spatial coordinates, were saved as .csv files for statistical analysis in R. Using the correlation of cell level DAPI signal from each staining/imaging round, a quality score was generated for every cell in each image, which ranges from 0–1 (0 being no registration, up to 1 for perfect registration). Only cells with quality score above 0.85 were included in the statistical analysis. Scores below 0.5 are generally due to tissue shifting/movement and loss. A summary of the minimum and maximum number of cells for each field of view (FOV) is shown in **S4A & S4B Fig**. Excellent correlations were found for the number of cells in the replicate slides for each cohort (>0.98). Slightly greater cell heterogeneity was found for one of the recurrent GBM slides but slide to slide correlation was still high (0.74) **S4C Fig**.

## Identification of cell clusters and biomarker co-expression

After exclusion of biomarkers which failed QC or staining criteria (as described above), unsupervised cell clustering was performed with subcellular data from 24 biomarkers (subcellular compartments used are indicated in **S3A Fig**). In total, 85,767 cells (from 20 treatment-naïve glioma cases) and 56,304 cells (from 16 recurrent GBM cases) were used to cluster cellular data for all biomarkers. Clustering was also conducted with smaller subsets of markers representing individual hallmarks (e.g. angiogenesis, invasion and energetics). Log2-transformed median cell intensity for each marker was used for K-means clustering. After trimming to reduce the impact of extreme outliers at both 2.5% tails, median cell biomarker values were standardized by the overall marker mean and standard deviation (since the distribution of marker intensity/expression values varied significantly within and between marker type).

Cells were clustered into K groups based on the multi-dimensional marker space (equivalent to number of markers used for clustering). The *kmeans* function provided by *stat* package of *R* (v. 3.4.1) was used with K (= 2 to 15). We used 10 random starts (nstart = 10) to address K-means clustering algorithm's sensitivity to initial seeds. We also used multiple metrics to determine the best number of clusters for the data such as Silhouette width, Calinsky criterion, Sum of squares of errors, and consensus clustering metrics. For consensus clustering (R ConsensusClusterPlus package), a subset of 5,000 randomly selected cells (due to computational constraints) were used. Consensus clustering iterates the clustering algorithm and examines if each pair of samples consistently clusters together or not. K-means clustering with Euclidean distance as metric was used for 1,000 iterations with 80% resampling. The cumulative distribution function (CDF) plot and the heatmap from consensus clustering were evaluated to guide us to determine the best number of clusters, aided by other metrics mentioned above. For a given K, each cell was assigned to one of the K clusters, and each tumor sample represented according to the proportion of cells belonging to one of the K clusters. For the purpose of visualization, lollipop plots were generated for each cluster and show the average protein profile of each cluster compared to the population average. Lines to the left of the vertical axis show lower than average expression, while to the right show higher than average expression. The percentage of cells belonging to each cluster was also calculated and shown for each lollipop plot.

## Exome and RNA Sequencing

Tumor and normal whole-exome sequencing and tumor RNA-sequencing data from the 20 treatment-naïve gliomas was studied; data was either produced from fresh-frozen tissue (n = 16, 8 of which had been sequenced in The Cancer Genome Atlas) or from FFPE tissue (n = 4) (**S1 Table**). Twelve of these were newly accessed for *de novo* analysis, and the remaining data was already available. Pathology estimates suggested those 12 samples all had greater than 70% tumor cell density and less than 50% necrosis. Data from sixteen post-treatment recurrent fresh-frozen glioblastoma tumors previously sequenced as part of a clinical trial [33] (data available in the database of Genotypes and Phenotypes (dbGaP) under accession number phs001460.v1.p1) was also included (**S2 Table**). All 16 of these tumors had whole-exome sequencing data, and fourteen had cognate RNA-sequencing data available.

Constitutional DNA from PBMCs was available for all 36 samples. For the eight fresh frozen glioma samples, Qiagen AllPrep DNA/RNA Mini Kit (cat#80204) was used to isolate DNA and RNA; for the four FFPE treatment-naïve glioma samples, Qiagen AllPrep DNA/RNA FFPE Kit (cat# 80234) was used. Exome libraries were constructed from 200ng of DNA (DIN = 3–5 for FFPE samples, DIN >8 for blood and fresh frozen samples) using KAPA Biosystems' Hyper Prep Kit (cat#KK8504) and Agilent's SureSelectXT V5 baits, containing custom content, following the manufacturer's protocols. Custom bait content included copy number probes distributed across the entire genome, along with additional probes targeting tumor suppressor genes and genes involved in common cancer translocations to enable structural analysis. For high quality RNA (RIN>6.0, DV200>90%), RNA libraries were constructed using Illumina's TruSeq RNA Library Preparation Kit V2 (cat#RS-122-2001) with 500ng inputs. For remaining RNAs (RIN<6, DV200>30%), libraries were prepared using Illumina's TruSeq RNA Access Library Prep Kit (cat#RS-301-2001) with either 40ng or 100ng inputs following the manufacturer's protocol and sample quality/input recommendations. Libraries were equimolarly pooled, quantitated, and sequenced by synthesis on the Illumina HiSeq 4000 for paired 82bp reads. FASTQ were aligned using bwa-mem (version 0.7.8) to the reference genome from 1000 Genomes project build hs37d5 with decoy contigs [b37d5] and Ensembl v74 for annotations. Somatic variants were called using lumosVar2 [38]. For this study, a tumor-normal mode was used which the sample fraction of clonal variant groups is set to zero in the constitutional sample.

## Deconvolution of samples into cell classes from RNAseq data of bulk samples

Multiple cell classes, characterized by different dominant biological processes, can be discerned by computational deconvolution of bulk gene expression data obtained from complex samples [39, 40]. This approach is a practical alternative when available samples are not suitable or available for single-cell sequencing (scRNAseq). Deconvolution assumes that the analyzed sample is composed of a certain number of cell types or different cell states, called *classes*. These classes do not necessarily fall into mutually-exclusive cell types. Instead, they represent quantifiable components of the analyzed samples that exhibit distinct gene- or pathway-attributable behaviors. We employed the previously published CellDistinguisher algorithm to identify sets of genes that are expressed predominantly in one class relative to the others [41]. As demonstrated in the Results, gene sets of ~50 genes led to robust assignments of cells into three classes. These distinguisher gene sets were then used to derive class signatures and compute sample compositions (fractions of cell types or classes in each sample) using the SSKL (Semi-supervised NMF algorithm for KL divergence) algorithm from the CellMix package [42]. To validate and support our findings with the multiplexed single cell data, we also

explored how well cell type assignments based on gene expression data compared to those based on protein expression measured by MxIF.

## Calculation of molecular and spatial cell heterogeneity metrics

Molecular and spatial heterogeneity metrics were computed for the MxIF spatially resolved cell data using a previously published heterogeneity analysis algorithm (MOHA) [43]. As described in more detail below, this technique computes the molecular "state" of each cell in a tissue section based on the fluorescence intensity of proteins within a given pathway, gene set or cancer hallmark [22]. Spatial "states" is a summated score which depicts the degree to which adjacent cells are of the same molecular state. The MOHA algorithm computes heterogeneity (or similarity or divergent states) metrics based on the distributions of these molecular and spatially defined states.

The molecular state of a given cell was defined as an ordered set of the values for each individual marker. A complete list of the cancer hallmark gene sets and the markers that were assigned to them is shown in **S3 Table**. The state of each marker was quantized into an ordinal value representing either a high, medium or low state, using the $33^{rd}$ and $67^{th}$ quantiles as the thresholds. The specific ordering of the markers in a given gene set (i.e. concatenation sequence) is arbitrary but was maintained consistently throughout the analysis. This process of computing the molecular state was repeated for each cancer hallmark marker set and for each cell. Next, molecular heterogeneity metrics were computed as a normalized Shannon's entropy of molecular states:

$$Molecular\ Heterogeneity = \frac{-\sum_{i=1}^{Nm} Pm_i \ln(Pm_i)}{\ln(Nm)}$$

The $Pm_i$ is the fraction of cells in molecular state i, and Nm is the number of possible molecular states in the system. The number of possible states for a gene set was defined as three raised to the power of the number of markers assigned to the gene set (e.g. 3^number of markers). The molecular heterogeneity metric value can range from zero to unity (i.e. maximum heterogeneity). For each patient tissue sample, a molecular heterogeneity metric was computed for each cancer hallmark.

Cell Spatial Heterogeneity is a summated score which depicts the degree to which adjacent cells are of the same molecular state as that of an index cell, with each cell in the tissue section serving as an index cell (example shown in **S5 Fig**). Identifying neighboring cells is necessary for computing the spatial heterogeneity metrics. Two cells were classified as neighbors if the Euclidean distance between the centers of the two cells was less than 1.3 times the sum of their radii. The cell radii were computed from the segmented cell area after approximating the cell as a circle. The spatial state metric was computed by surveying the neighbors of each cell and counting only the number of neighbors in the same molecular state. This number of neighbors represents the cell spatial state for each pathway or gene set. Having no neighbors in the same molecular state is a valid cell spatial state. Therefore, the cell spatial state can range from zero to the maximum number of neighbors a cell has. After going through every cell and their neighbors, a frequency distribution was established for these cell spatial states. The cell spatial heterogeneity was then computed as a normalized Shannon's entropy of spatial states:

$$Cell\ Spatial\ Heterogeneity = \frac{-\sum_{k=0}^{Z_{max}} Ps_k \ln(Ps_k)}{\ln(Z_{max}+1)}$$

where, $Ps_k$ is the probability of state k, and $Z_{max}$ is the maximum number of neighbors a cell can have as measured in the tissue sample. For each patient tissue sample, a spatial heterogeneity metric was computed for each cancer hallmark.

## MRI imaging protocols and image feature extraction

The multiparametric MRI (mpMRI) exams of the brain consisted of T2-weighted (T2), T1 weighted pre-contrast (T1 Pre), T1 weighted post contrast (T1 Post), Apparent Diffusion Coefficient (ADC) derived from diffusion-weighted imaging (DWI), and Fluid Attenuated Inversion Recovery (FLAIR) images. The subjects with recurrent GBM were imaged using 3 Tesla GE scanners, while the treatment naïve subjects were imaged at a different institution using 3 Tesla Siemens scanners. Although the acquisitions were consistent in sequence types across institutions, parameters such as relaxation and echo times were different, thus prompting separate image analysis for the two cohorts.

Tumor annotations on the MR images were manually outlined by an expert neuroradiologist to depict the extent of the whole tumor, including peritumoral regions, relative to the FLAIR sequence. To the extent possible, an equivalent normal region on the contra-lateral side of the brain was demarcated. A deep learning approach was trained on the Brain Tumor Segmentation (BraTS) challenge data [44] and was utilized to divide the whole tumor segmentation into enhancing core and necrotic core based on T1-post contrast MRI. A U-net network was trained using the T1 Post contrast MRI to identify the extent of the enhancing and necrotic cores on the BraTS data. The training code and trained model are available (https://github.com/mirabelarusu/deep_learning_inference_browser). The trained model was subsequently applied on the T1 post contrast MR images for the patients in our cohort to segment the enhancing and necrotic cores. The peritumoral (edema) regions were obtained by subtracting the enhancing and necrotic core from the whole tumor segmentation. Manual corrections and automatic postprocessing were utilized when appropriate to improve the precision of the annotations or remove minor disconnected regions. At the completion of these processing steps, an annotation of the whole tumor, the peritumoral (edema) region, enhancing core, and necrosis were obtained for each subject relative to the FLAIR protocol.

Pre-processing steps were applied on the mpMRI prior to feature extraction, including spatial registration to align the FLAIR protocol relative to the others, in order to project the region annotations on the rest of the protocols. Intensity normalization was applied in the entire organ by using the normal regions as reference. Specifically, the intensities were normalized such that the average intensity in the normal region had a value of 1. To perform this normalization, we divided the intensity of each voxel by the average of intensities within the normal region.

Image derived quantitative features were evaluated for each subject. Due to the limited number of subjects in our study, the large number of protocols (n = 5) available for each subject and the multiple subregions available for each tumor (n = 4), we chose to consider only three protocols (T1-post, FLAIR and ADC) and three tumor subregions (the whole tumor, the peritumoral edema and enhancing core). We represented the tumor subregions by two image-derived quantitative features (mean and standard deviation), resulting in 18 image-derived features per subject. Also, for each subject, we included three morphologic features (the volume of the enhancing core, the volume of the entire tumor and their ratio–which we refer to as the normalized enhancing core volume).

## Multimodality data integration and clustering

Finally, we investigated the associations between imaging quantitative features and other variables including cell cluster data, clinical parameters and cancer hallmarks based on cell protein expression, RNA and DNA. Due to the different source and scales of the multimodal data (clinical, MxIF, genomic, MRI), we discretized the most relevant features into "low", "medium" and "high" groups, based on the data ranges across the individual cohorts. Features

were considered to be relevant for the multimodal association analysis either because they were clinically utilized for decision making, e.g. IDH1 mutation status, age, and grade, or because they showed consistent trends across both treatment naïve subjects as well as recurrent GBM subjects. Based on the discretized variables, subjects were then clustered using hierarchical clustering with the Euclidean distance metrics.

## Statistical analysis and data visualization

Using R software [45], individual biomarker expression differences between IDHmt and wt patients were calculated using unpaired Student's *t*-tests, and multivariate Random Forest classifier was used to classify IDHmt vs IDHwt patients using those features identified in the *t*-test, with the performance of the classifier reported in both error (out-of-bag error rate) and AUC (leave-one-out cross-validation). The proportions of cell clusters between IDHmt and wt tumors were also compared using unpaired *t*-test. The p-values were not adjusted for multiple testing.

R was also used to generate plots and figures for visualizing the molecular and spatial heterogeneity. The Mann-Whitney test was performed on all unpaired comparisons of molecular and spatial heterogeneity metrics by condition with the computed p-value reported in each plot. The p-values were not adjusted for multiple testing. The Pearson correlation coefficient was computed and included on each of the slide to slide correlation plots.

We compared the MRI-derived features for patients that are IDHwt or IDHmt using unpaired *t*-tests. False discovery rate was used to control for multiple comparisons. Similarly, we compared the MRI-derived features within the binned RNA and protein expression levels using unpaired *t*-tests across each bin pair within the three groups. Multiple comparison correction was applied within each MRI-variable evaluation (i.e. 3 comparisons per panel).

For survival analysis, Cox Proportional Hazard model was used for comparing IDHmt vs wt patients, and Kaplan-Meyer curve was used to plot survival rates.

For the purposes of data visualization and interpretation, data was aligned by cluster, IDH1 mutation and patient ID. Biomarker intensities were grouped by cancer hallmarks (invasion; energy metabolism; angiogenesis; stem cells; immune response; proliferation; resisting cell death; DNA damage) and iron metabolism.

## Results

### Marker expression differences between IDH1 mt and wt tumors

Univariate and multivariate analysis of biomarker expression in the treatment-naïve glioma cohort showed significantly lower values for vimentin (p = 0.0002), VEGFR2 (p = 0.0002), nestin (p = 0.003), Ki67 (p = 0.006) and HLA1 (p = 0.008) proteins in IDHmt vs wt tumors (**S6A Fig**). Three of these, VEGFR2, vimentin and HLA1 were also included in the multivariate model using Random Forest which provided an AUC of 0.87 (error rate 5%) in predicting IDH mutation status (**S6B Fig**). Since a majority of IDHmt tumors are derived from oligodendrogliomas which minimally express vimentin [46, 47] and IDHmt tumors are known to have suppressed angiogenic pathways, differential expression of VEGFR2 and vimentin between IDHmt and IDHwt is not surprising.

### Cellular and genomic analysis shows cancer hallmark differences in IDH1 mt vs wt tumors

**Cellular differences in IDHmt vs wt tumors.** In total, 24 markers across 85,767 cells from the 20 treatment-naive glioma cases underwent k-means clustering. **Fig 2** shows

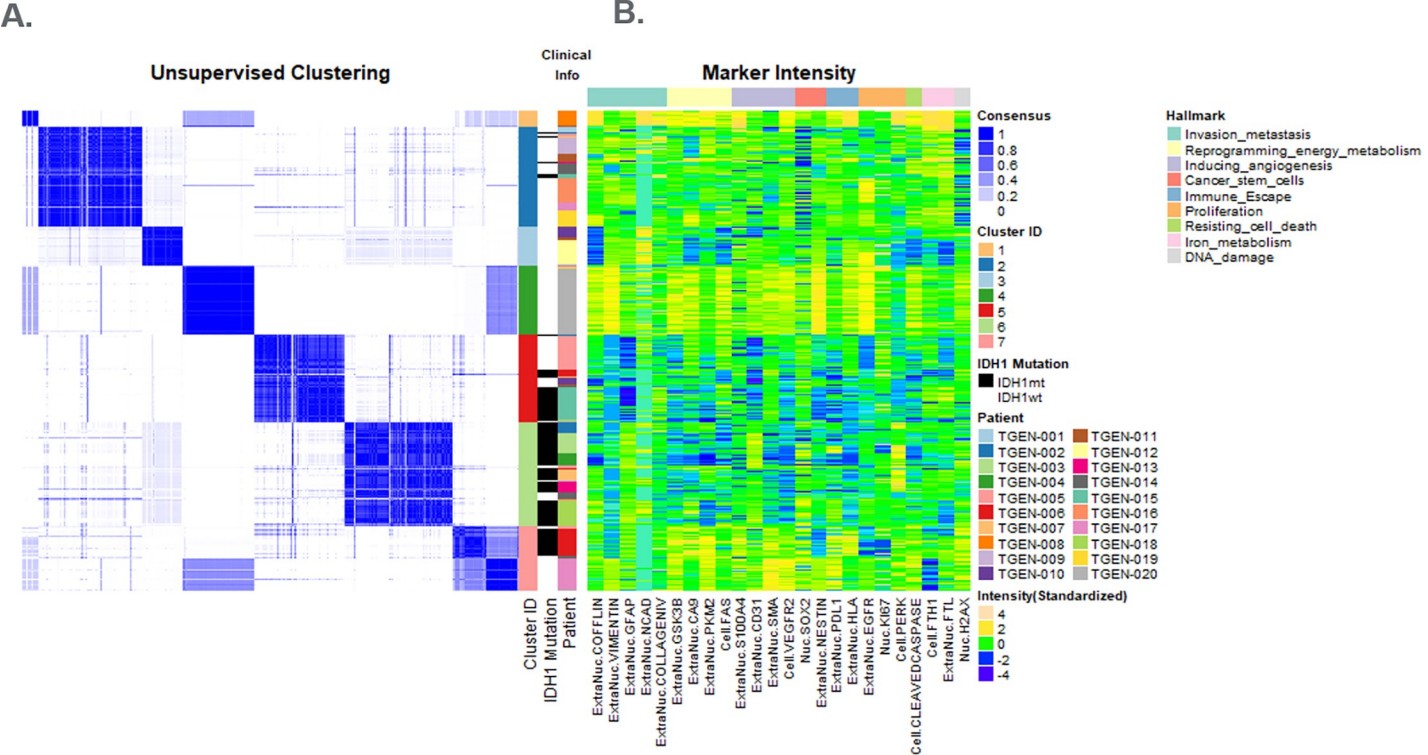

**Fig 2. Distribution and clustering of cells based on protein expression from all treatment-naïve patients.** Unsupervised clustering of cell biomarker data revealed 7 distinct subsets (clusters) of cells. Cluster 2 is dominated by IDH1wt and Cluster 6 is dominated by IDH1mt cases. Clusters 1, 4 and 7 (which were less diverse patient groups) show higher staining intensities of most markers (and cancer hallmarks) compared to Clusters 2, 5 and 6. Iron metabolism marker expression was generally high in Cluster 2, but low in Cluster 6.

unsupervised clustering and segregation of the cells into 7 clusters; marker intensity organized by cluster, IDH1 mutation and cancer hallmarks (invasion; energy metabolism; angiogenesis; stem cells; immune response; proliferation; resisting cell death; DNA damage) and iron metabolism. Relative biomarker intensities (compared to population mean) for each cluster are shown in the lollipop plots in **S7 Fig** and were evaluated visually. Clusters 1 and 4 were composed of cells from just two IDH1wt patient cases and had higher than average expression of most proteins (and hence hallmarks) (**Fig 2**). Clusters 2 and 6 contained the largest numbers of cells (21.0% and 21.9%, respectively, **S7 Fig**) from the greatest number of cases (12 and 11 cases, respectively, **Fig 2**). Cluster 2 was predominantly composed of cells from IDH1wt tumors, while cluster 6 was predominantly composed of cells from IDHmt cases. Cluster 2 lollipop plot shows lower expression of γH2AX, Sox2, SMA, and Ncad and higher expression of FTL and FTH1, while remaining proteins were close to the population average. Cluster 6 had lower expression of most proteins (**S7 Fig**), except pERK, CD31 and Ncad. Cluster 5 had average or lower than average expression of all proteins. Cluster 7 had average or above average expression for most proteins. Both clusters were comprised of cells from both IDHwt and IDHmt cases. Notably, visual inspection of the lollipop plots showed that both angiogenesis and metabolism-related markers were generally lower in IDH1mt cases, as was expression of antigen presenting machinery, i.e. HLA1, and invasion markers collagen IV and vimentin (consistent with our earlier analysis). Interestingly, IDHwt cells had higher expression of ferritin light and heavy chains, indicating increased iron storage in these cells. Removal of free iron by enhanced iron storage has been implicated in evading ferroptosis by cancer cells. A more

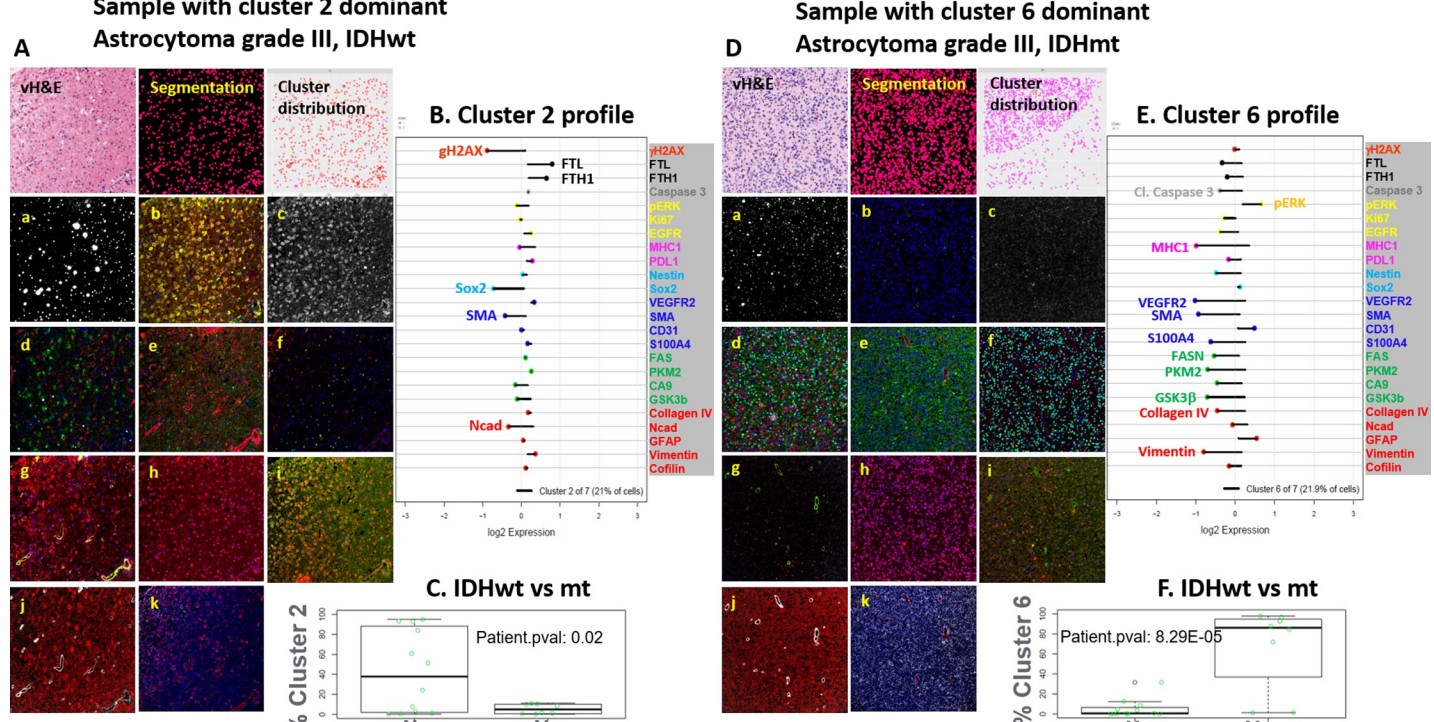

**Fig 3. Biomarker images and lollipop plots for cluster 2 and cluster 6.** Staining images for representative cases in Cluster 2 (A) and Cluster 6 (D), including a vH&E image (top left), segmented image (top middle) showing individual cells, an image with cluster assignment to individual cells (top right) and a number of single marker or multi-marker overlays representing expression of different hallmark proteins ((a): DNA breaks, gH2AX. (b): Iron metabolism; FTL, FTH1; (c): Cell Death, Cleaved Caspase 3; (d): Proliferation, EGFR, pERK, Ki67; (e): Immune MHC1, PDL1; (f): Stemness, Nestin, SOX2; (g): Angiogenesis, VEGFR2, SMA, S100A4, CD31; (h) Metabolism, FASN overlaid on DAPI (i) Metabolism, GSK3b, PKM2, CA9; (j) Invasion, GFAP, Collagen IV; (k) Vimentin, Cofilin & NCad.Panel B and Panel E show the protein expression profiles of individual clusters (2 & 6, respectively); "lollipop" lines originate at the average expression of proteins in all cells measured from all cases. Lines to the left of the average vertical axis show lower than average expression, while to the right show higher than average expression. Cluster 2 (Panel C) and cluster 6 (Panel F) trend towards separating cases by IDH1 mutation status. Specifically, Cluster 6, which shows a lower than average expression of most hallmark proteins, is significantly positively correlated to IDH1 mutation (Panel F); Cluster 2 cells with higher iron metabolism (FTL, FTH1) show a trend towards lower representation in IDH1 mutant samples (Panel C).

in-depth analysis of this pathway is necessary to determine if evasion of ferroptosis is indeed driving the tumor growth in these patients. **Fig 3** shows two representative examples of IDHwt and mt tumor samples, with biomarker staining examples and lollipop plots for proteins in clusters 2 and 6. IDHmt patients had significantly lower proportion of cluster 2 cells (**Fig 3C**; p = 0.02) and higher proportion of cluster 6 cells (**Fig 3F**; p<0.0001 vs IDHwt patients). Overall, similar staining profiles and cluster patterns in IDHmt vs wt cases were found in the recurrent GBM cohort (data not shown).

Clustering of cells by angiogenesis, invasion and reprogramming cellular energetics hallmark proteins was also conducted. Confirming earlier results, IDHmt tumors had a significantly higher proportion of cells with low expression of angiogenesis markers vs IDHwt tumors (p<0.001). A summary heat map of angiogenesis clusters and IDHmt and wt patients shown in **S8 Fig**. Otherwise, for the other hallmarks, IDHmt patients had higher proportion of clusters with low invasion and energetic marker expression.

**Cell cluster alignment with IDH and other glioma related mutations.** **Fig 4** shows cell cluster distribution (**A**) aligned with IDH mutation status (**B**) and the other most common mutations in treatment-naïve glioma patients. In concordance with known biology, IDH1 mutations were found to be mutually exclusive of EGFR and PTEN mutations (**Fig 4B**).

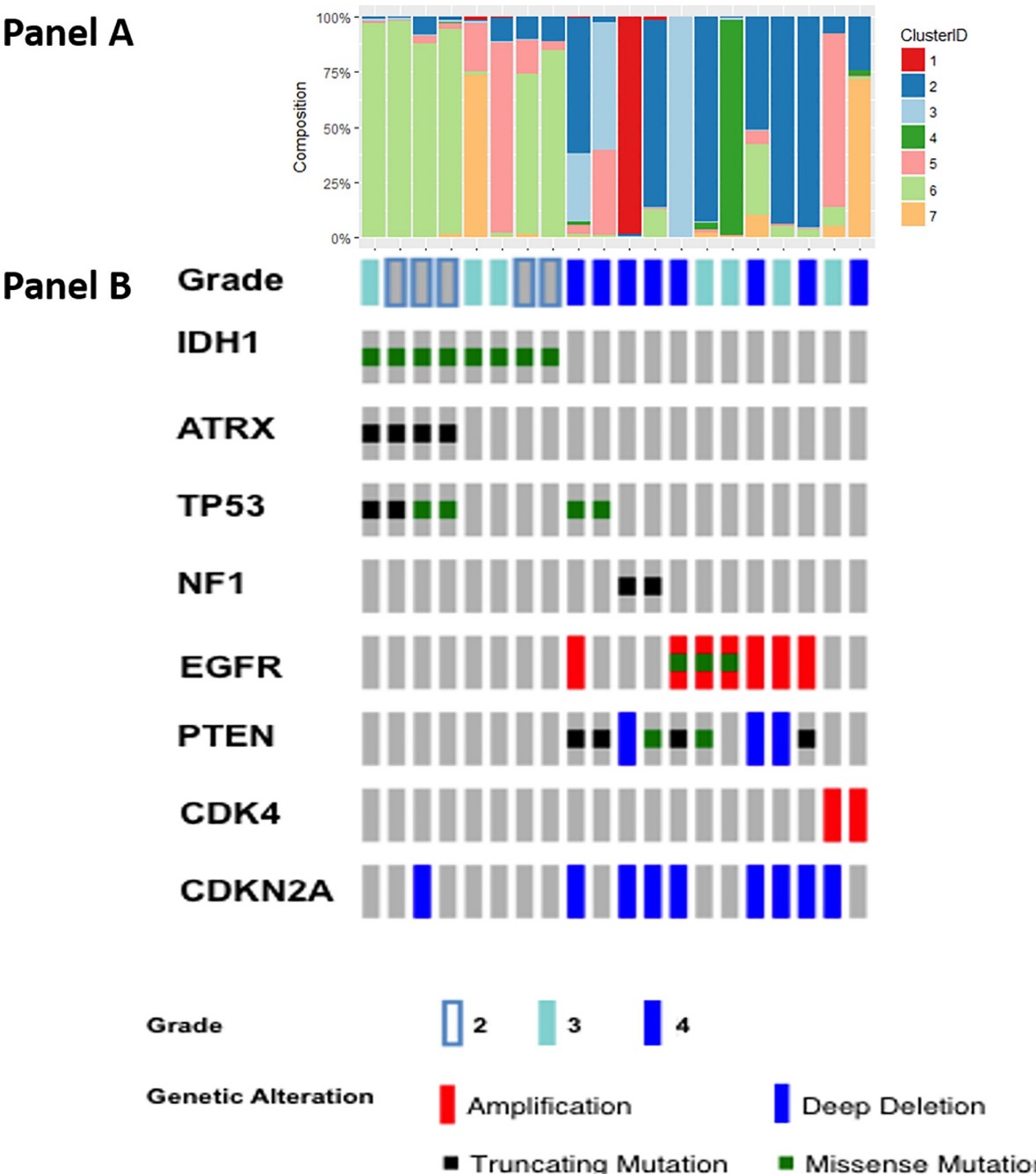

**Fig 4. Cell cluster composition and Oncoprint of treatment naive gliomas.** For each glioma case, Panel A portrays the fractional distribution of its cells within each of the 7 clusters. Panel B depicts the genomic profile of each glioma case.

IDH1mt samples appeared to be more homogenous, particularly those with concurrent ATRX mutation, and were mostly dominated by the cluster 6 cell phenotype (based on earlier analysis had lower than average expression of most markers). Approx. 50% of IDH1wt cases with EGFR amplification had a high proportion of cluster 2 cells (overall, average biomarker expression, and lower DNA damage and stem cell markers, higher iron metabolism markers).

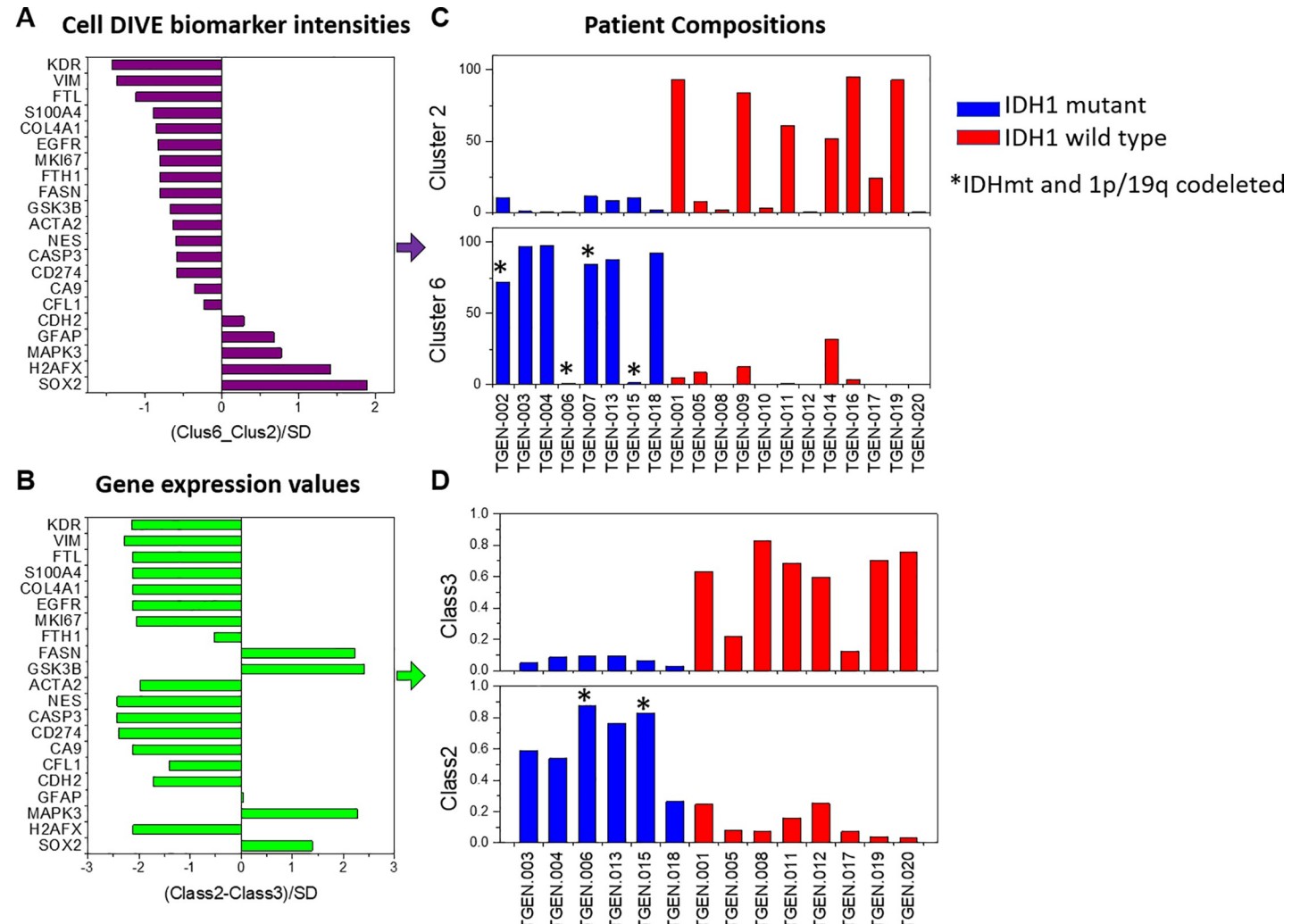

**Fig 5. IDH1 mutation status drives cell phenotype at both the gene and the protein level.** Ratios of the average staining intensities for 21 MxIF markers in clusters 6 and 2 were calculated (Panel A). Following deconvolution of the transcriptomes using CellDistinguisher, RNA expression counts (FPKM) for the mRNAs were used to distinguish "class types" (n = 3) across the bulk sequenced specimens, then ratios of the expression values for the same 21 genes compared between Class 2 and Class 3 (Panel B). Fractional composition of each patient case by Cluster 2 or Cluster 6 (Panel C) or according to Class 2 or Class 3 (Panel D) was determined. Cases dominated by cells belonging to protein cluster 2 were more likely to be found in IDH1 wild-type tumors, while cases for which cells from cluster 6 dominated were mostly IDH1 mutated tumors (Panel C). Similarly, the fractional composition of glioma cases comprised of gene expression class 3 were present in higher proportions in IDH1 wild type samples, while class 2 cell types were more abundant in the IDH1 mutant ones (Panel D). The distinguisher genes of class 3 were enriched in genes related to cancer hallmarks of "inducing angiogenesis", "enabling replicative immortality" and "evading growth suppression" (see **S4 Fig** and **S2 Table**).

**Cell cluster alignment with RNA expression and IDH status.** The degree to which protein-based single cell clusters concur with deconvoluted, transcript-based cell class assignments across treatment-naïve gliomas with IDHmt or wt was evaluated. Based on the gene expression data of all measured genes, we identified three cell classes using CellDistinguisher, each class having 50 or more distinguisher genes (**S9 Fig**). Exceeding three classes resulted in a very short list of distinguisher genes for some classes, which diminishes the utility of comparing states or functions across the classes. Class 2 and Class 3 were qualitatively similar to protein derived cell cluster 6 and Cluster 2, respectively (**Fig 5**). Ratios of the average staining intensities for 21 markers in clusters 6 and 2 were calculated (**Fig 5A**). The ratios of the expression values for the same 21 genes were compared between RNA classes 2 and 3 (**Fig 5B**).

Fractional composition of IDHmt and wt cases within cell cluster 2 or 6 (**Fig 5C**) or within RNA class 2 or 3 (**Fig 5D**) was determined. Consistent with earlier results, tumors dominated by cluster 2 cells were more likely to be IDHwt, while cases with dominance of cluster 6 were mostly IDH1mt. Similarly, the IDHwt tumors were mainly comprised of RNA class 3 markers while class 2 was more abundant in the IDH1mt (**Fig 5D**). IDH1wt tumors were enriched in class 3 cells (enriched in genes related to the cancer hallmarks of inducing angiogenesis, enabling replicative immortality and evading growth suppression), while the IDH1mt samples had a lower abundance of genes related to these cancer hallmarks.

Despite the differences in sample amount and preservation (fixed vs frozen), we have found noteworthy similarity between the cell types and patient compositions identified from the protein biomarker intensities and the gene expression data. Except for FASN, GSK3b and NCad, good directional correlation was observed in differential protein and gene expression between cell clusters and RNA classes in the IDH1mt and IDHwt populations (**Fig 5**). Lack of concordance between H2AX protein and transcript likely is due to staining intensity by anti-γH2AX antibody reporting only the post-translationally phosphorylated form of the protein (instead of total protein, which the transcript count would more reasonable reflect). It is unclear why there was not good concordance for FASN, GSK3b and NCad, but directional discordance between mRNA and protein for any given gene is not an unanticipated finding, and on an individual gene level, is more discordant in tumor versus cognate normal tissue [48]. Underlying reasons for this discordance include variant turnover rate of the protein versus mRNA, translational efficiency due to ribosomal density and occupancy, RNA secondary structure, among others [49]. However, the high concordant directionality of 17 of the 21 markers argues for robustness of the biological inference that molecular features in cells from treatment-naïve gliomas are related to IDH1 mutation status. We conclude that biomarker-based clusters 6 and 2 refer to the same cells and/or processes as gene-expression-based classes 2 and 3. Although at individual gene levels, mRNA and protein expression values don't evince quantitative direct, strong correlation, our findings indicate that looking at the behavior of cells at the gene set or pathway level can lead to consistent patterns starting from different data types [49, 50].

**Intratumor and spatial heterogeneity.** In addition to the cell level protein expression and cell composition within the IDHmt and wt tumors, we further investigated molecular and spatial heterogeneity of the biomarkers in each of the hallmark categories. Examples of the heterogeneity metrics for the cell proliferation hallmark (comprising Ki67, nestin and EGFR) in gliomas and recurrent GBMs are shown in **Fig 6A,** which shows the discretized (high (2), medium (1), low (0)) expression values for each marker, and corresponding color-coding for each cell. Heterogeneity calculated from the distribution of these states in different tumors shows an inverse correlation between molecular and spatial heterogeneity in both treatment-naïve glioma and recurrent GBM cohorts. IDHwt tumors had higher molecular heterogeneity while IDHmt tumors were more spatially heterogeneous (**S10 Fig**). Similar trends were present in both cohorts. **Fig 6B** shows a scatter plot of heterogeneity in the "inducing angiogenesis" hallmark with the range of spatial and molecular heterogeneity metrics for gliomas and recurrent GBM samples, also encoded by IDHmt (red) and wt (blue) status. Trends in heterogeneity of this hallmark were similar to those observed for the proliferation hallmarks, as well as activating invasion motility hallmark (S10 Fig). No other significant differences in heterogeneity were found.

## MR feature differences between IDH1 mutant and wildtype patients

Simple features derived from the MR images uncovered differences in discernable elements of brain tumor dispersion from IDH1wt and IDH1mt patients. IDH1wt patients had larger

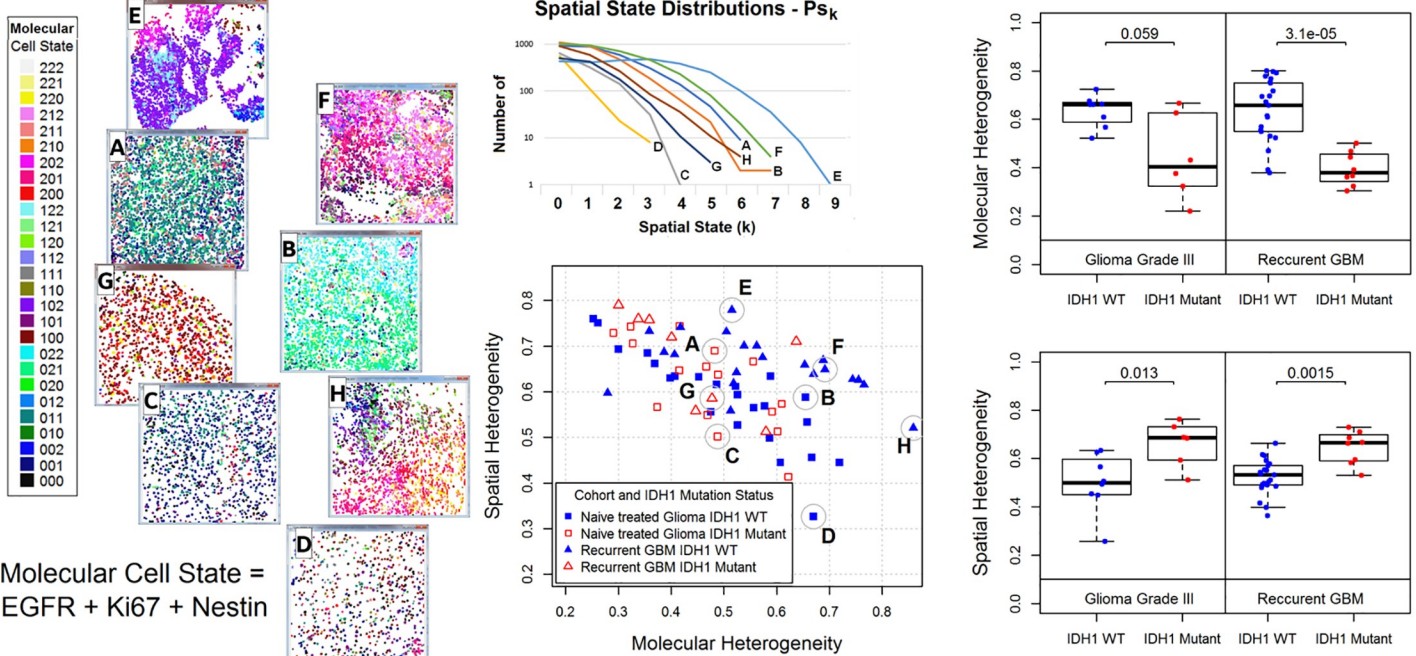

**Fig 6. Computed molecular and spatial heterogeneity metrics using the multi-omics heterogeneity analysis (MOHA) tool.** The method first converts the continuous marker intensity measures of each segmented cell into an ordinal value representing either a high, medium, or low state. Panel (a) presents an example for the Sustaining Proliferative Signaling cancer hallmark. This gene set is composed of three markers: EGFR, Ki67, Nestin. The state of each of these markers can either be high (2), medium (1), or low (0). Therefore, the three-marker gene set has 27 possible molecular states presented in the color-coded legend (far left). The scatter plot (center) presents the spatial and molecular heterogeneity of treatment naïve gliomas and recurrent GBM samples. Images of tissues from four treatment naïve gliomas (A-D) and four recurrent GBM (E-H) are presented with each segmented cell colored by their expressed molecular state. The spatial state distributions of these eight samples are presented above the scatter plot. For the 4-gene set "inducing angiogenesis" (SMA [ACTA2], VEGFR2 [KDR], CD31 [PECAM1], and S100A4) hallmark, IDH1 mutation status discriminates those cases with relatively lower molecular heterogeneity and relatively higher spatial heterogeneity in grade III treatment-naïve glioma or recurrent glioblastoma. Mann-Whitney test p-values are presented for each nonpaired comparison. (panel b).

enhancing cores (feature "Normalized enhancing core volume"), but less contrast uptake in the peritumoral edema regions (feature "Edema T1 post"). On the other hand, the IDH1mt patients lack a clearly defined enhancing core, but have increased contrast uptake on the T1 post contrast MRI protocol in the peritumoral edema region (**Fig 7**). These trends were observed both in the treatment-naïve glioma as well as the recurrent GBM, and are not surprising since the IDH1mt are known to have less contrast enhancement than the IDH1wt [51].

Other intensity and volumetric features were evaluated on clinically important MRI protocols, e.g. ADC or FLAIR, but they failed to show separation between IDH1 mutational status or a consistent trend across the two cohorts. Thus, our analysis focuses on the normalized enhancing core volume–measuring the enhancing core volume normalized to the entire tumor volume, and the T1w MRI post contrast uptake in the peritumoral edema region. Statistical significance was not achieved for any features after multiple comparison corrections likely due to the small number of patients in each cohort.

## Multimodal data association

Unlike previous studies [27],[28–31] that focused on predicting IDH1 mutational status using MRI features, we assessed the correlations of MRI features with genomic and proteomic markers within the angiogenesis hallmark to characterize the differences between IDH1 mutational

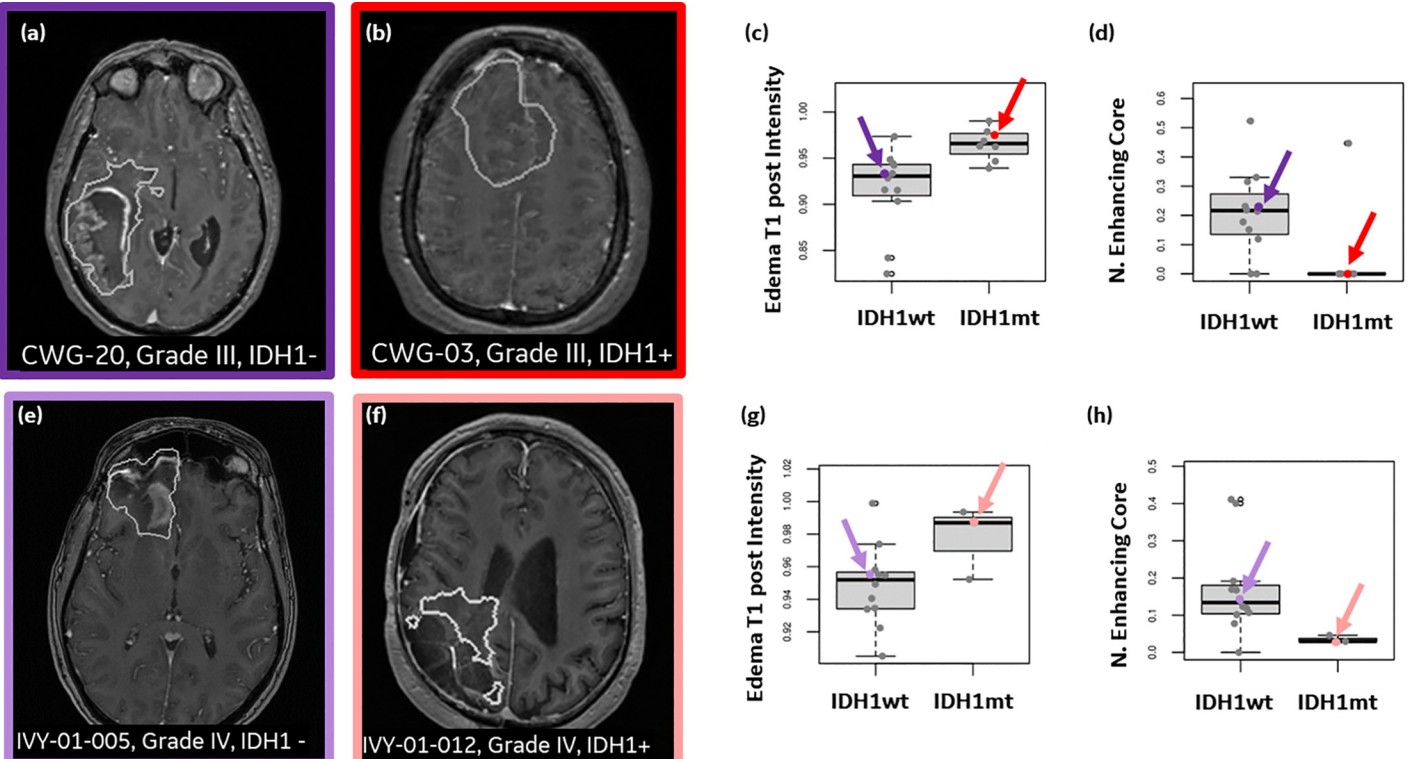

**Fig 7. MRI-derived features appear to differentiate patients that carry an IDH1 mutation (IDH1mt) and those that are wild type (IDH1wt), regardless if subjects are treatment naïve or have recurring GBM.** T1w post contrast MRI for IDH1wt subjects (A,E), and IDH1mt (B,F). The white outlines show the extent of the tumor as delineated by the expert neuroradiologist (LW). (C,G) Across the two cohorts, a similar trend may be notice when comparing the mean T1 post-contrast intensity signal in the peritumoral edema region, suggesting an increase in enhancement in the IDH1mt in the peritumoral edema region when compared to the IDH1wt (C and G). An opposite trend is observed when comparing the normalized enhancing core volume across IDH1wt and IDH1mt (D and H), indicating that IDHmt patients have limited to no enhancement. None of these comparisons reach statistical significance after multiple comparison correction using false discovery rate.

status. **S11 Fig** shows that larger enhancing cores are associated with higher RNA expression levels in the "inducing angiogenesis" hallmark. A similar association is observed with the expression levels of protein markers, i.e. S100A4 that is known to promote angiogenesis and metastasis development [52], and VGFR2 that plays a fundamental role in neovascularization [53]. These found associations were consistent regardless of the type of tumor, treatment naïve glioma or recurrent GBM.

When investigating multimodal associations (**Fig 8**), we can also observe a consistent trend across the two cohorts of patients. Not surprisingly, IDH1 mutations are found in lower grade tumors, younger patients and have better overall survival. As also shown in **Fig 7** and **S11 Fig**, IDH1mt tumors have smaller enhancing cores but more contrast uptake in the edema regions and show reduced expression levels of RNA and protein from the "inducing angiogenesis" hallmark (**Fig 8**, highlighted box). Of the five angiogenesis hallmark cell clusters, cluster 4 (above average expression of VEGFR2, SMA and CD31) and cluster 5 (above average expression of VEGFR2 and S100A4), which are characterized by higher expression of angiogenesis markers, show low cell percentages in the subjects with IDH1 mutations. On the other, the IDH1wt tumors are molecularly more diverse and show more heterogeneous multimodal variables, yet still a general trend of higher expression levels of RNA and protein markers involved in inducing angiogenesis and reduced overall survival. Clusters with average (cluster 3) and lower than average expression (clusters 1 and 2) were distributed among all patients, however,

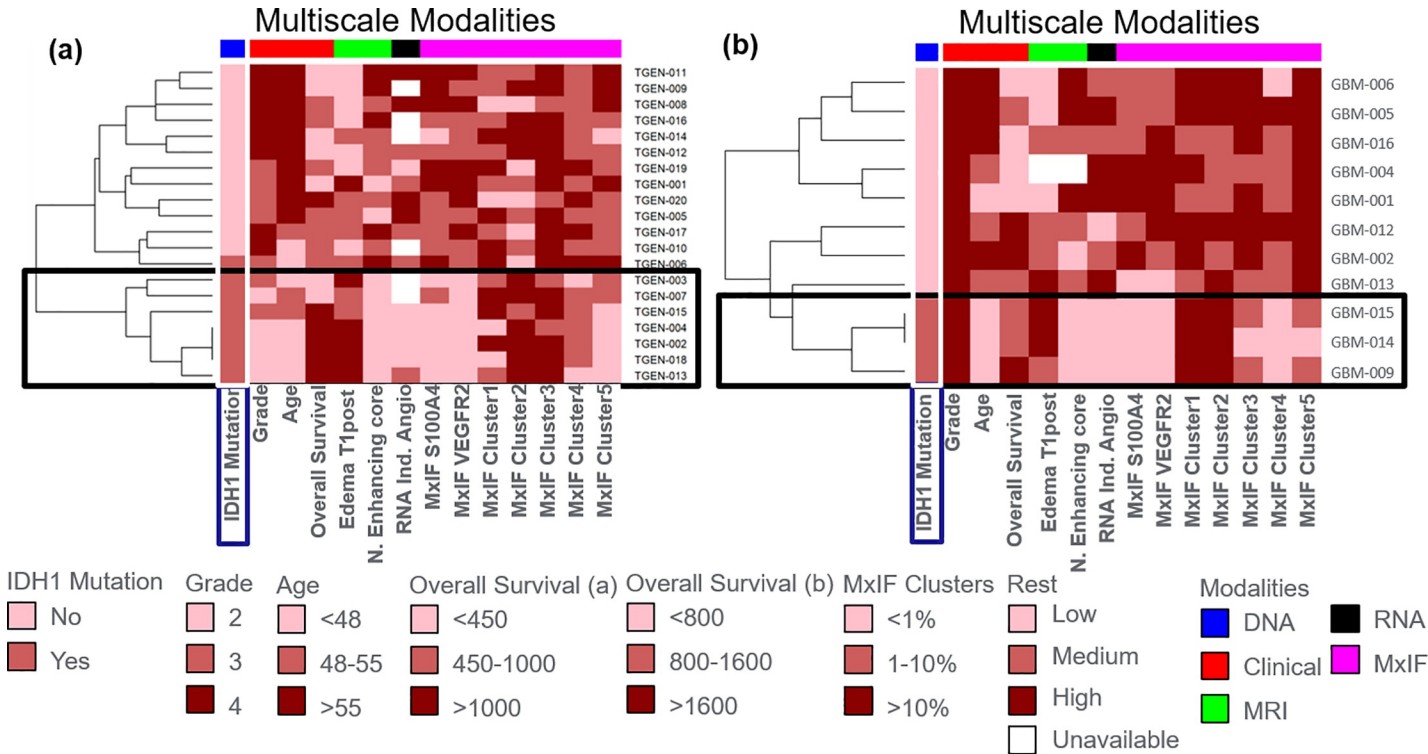

**Fig 8. Comprehensive rendering of multiscale measurements in gliomas.** Multiscale modalities depicted include: 1) clinical information (red), 2) IDH1 mutational status (blue), 3) MRI derived variables (green), 4) RNA expression level of genes involved in the "inducing angiogenesis" hallmark (black), and 5) MxIF angiogenesis markers or cell clusters (magenta). The data is binned in low, medium and high categories. Across the treatment-naïve gliomas (a) and the recurrent (post-treatment) glioblastoma* (b) cohorts, it can be observed that IDHmt patients have low angiogenesis according to RNA expression levels and expression of S100A4 and VEGRF. Those subjects also have high fraction of cells in clusters 1 and 2 (low angiogenesis markers), and low fraction of cells in cluster 4 and 5 (high angiogenesis markers (**S8 Fig**, cluster profiles of angiogenesis clusters). Moreover, MR Images for the same subjects have lower normalized enhancing cores volumes and measure higher intensities on T1 post contrast. *Recurrent GBM (5 subjects are not shown since they were missing MxIF).

relative proportion of these compared to the other two clusters was much higher in the IDHmt patients. Age, grade and histology are confounding factors in the recurrent GBM progression cohort as IDH1mt tumors tend to occur at younger age and are generally low grade oligoden-drogliomas, however, as the similar trends were apparent in the recurrent cohort, which are all grade IV GBMs, these observations probably reflect differences in biology between the IDH1mt and IDH1wt tumors.

## Discussion

We deployed a multiscale workflow which accommodates biomedical imaging (multi-parameter MR imaging) of glial tumors, *in situ* multiplex immune detection of discrete biochemical functional states in tissue sections from tumors at single cell level, and next generation sequencing of DNA and RNA from those same tumors. The data produced by each technology was post-processed to regions-of-interest and features (MRI), molecular state assignments of individual cells in tissue (based on gene sets and signaling pathways interrogated by specific antibodies), and molecular subtyping, pathway and hallmark mapping (determined by mutations and cellular deconvolution from bulk RNA sequencing). A coherent picture of enhanced angiogenesis in IDHwt tumors evident in non-invasive in vivo imaging features emerges from the data derived from multiple platforms (genomic, proteomic and imaging) and scales from individual proteins to cell clusters/states as well as bulk tumor. Results are consistent with

known observations at the molecular (suppression of proangiogenic markers in IDHmt tumors) and imaging scales (no or low enhancement in IDHmt tumor), but now fill in the gaps on how the two are linked through the intermediate scales of cellular states and their spatial organization. Multiplexed immunofluorescence (MxIF) staining using 43 antibodies on individual tissue sections (duplicate punches in a tissue microarray) afforded insight into the clustering of single cell functional states from 20 treatment-naïve gliomas (grades 2–4) into 7 clusters. Discreet patterns of protein abundance across 7 hallmark phenotypes and 2 biochemical signature events (iron metabolism and DNA damage) suggest that broad segregation of such functional states may be associated with IDH1 mutation status. Among the more robustly discriminating hallmarks between IDH1 wildtype from IDH1mutant gliomas is that of angiogenesis. The enhancement patterns, specifically how much of the tumor enhances (assessed by the "normalized enhancing core volume" feature) and the contrast uptake in the peritumoral edema region (Edema T1 post intensity), appear to be consistently correlated with the IDH1 mutational status, a trend that is conserved across the two independent cohorts we investigated. Our findings suggest that the IDH1wt tumors have a more consistent enhancing pattern with a clearly defined enhancing rim and little uptake elsewhere. On the other hand, the IDH1mt tumors have a diffuse appearance on MRI without a well-defined enhancing rim and with higher uptake in the edema region, on account of infiltrating cells. Previous studies have linked poor survival with the peritumoral edema volume [54] and tumor volume [27]. Moreover, IDH1mt tumors are known to have less edema [51]. From the richness of the molecular heterogeneity portrayed from MxIF scoring, comparing the functional states of adjacent cells (whether they are similar or dissimilar) affords a calculation of spatial heterogeneity across the different hallmark phenotypes. Here we find the unanticipated segregation of both treatment-naïve gliomas as well as recurrent glioblastoma based on IDH1 mutation status within hallmarks of "invasion motility", "proliferative signaling", and "inducing angiogenesis". The genomic profiling depicted what is already known about glial tumors, (the mutual exclusivity of IDH1 mutations with EGFR and PTEN mutations, the co-existence of ATRX mutations only within a subset of IDH1 low grade tumors, etc), but also revealed the heretofore unknown frequent, diminished molecular heterogeneity of IDH1mt low grade tumors. Removal of free iron by enhanced iron storage has been implicated in evading ferroptosis by cancer cells [34, 35]. Cluster 2, which was highly represented in IDH1wt tumors showed an increased expression of iron storage markers (FTL and FTH1) and decreased expression of γH2AX, a marker of DNA breaks (S7 Fig). This is consistent with increased sequestration of iron, making it unavailable for oxidative DNA damage leading to evasion of ferroptosis. A more in-depth analysis of this pathway that includes iron transport, storage and utilization is necessary to determine if evasion of ferroptosis is indeed driving the tumor growth in these patients [55, 56]. Inter- and intra- tumoral molecular heterogeneity is a well-recognized feature of GBM [6, 57, 58] and is believed to be the main reason behind treatment failure. Emergence of several single cell analysis platforms has fueled the investigations of intra-tumoral heterogeneity of glioma [17, 59–61], including tumor-stromal cell interactions [62, 63] as well as interactions between the diverse tumor cell populations [64, 65]. Importance of the intercellular interactions among heterogeneous tumor cell population is highlighted by the observations of Inda et. al. [64] that EGFRmt cells that are far outnumbered by the EGFRwt population drive enhanced proliferation of these cells by paracrine signaling thereby driving tumor growth. Thus, tools to evaluate molecular and spatial heterogeneity and cell-cell interactions are likely to unravel heretofore unknown mechanisms that drive tumor growth and/or treatment failure. IDH mutation induced suppression of immune response has also been noted previously, however, it has been linked to decreased expression of effector T cell response related genes [66]. Whether this in turn affects the expression of HLA1 in IDHmt tumors is not known.

## Study limitations

The key limitations of this study include small sample size, lack of registration of sample derived for molecular analysis to MR images and a limited number of markers representing different hallmarks. The intent of this study was not to generate a diagnostic signature but to evaluate correlation between imaging and molecular features at the hallmark level and to generate a workflow for integrating multiscale multiparametric data to study disease biology. While the sample size (n = 20) in the treatment-naïve glioma cohort was limited, the fact that similar cell clusters existed in another cohort (recurrent GBM) and the correlations between MR and molecular features of angiogenesis hallmark hold for both cohorts is encouraging. Having developed methods to integrate and evaluate such a complex data set, we are in the process of designing a more focused study to interrogate the biology of a specific molecular subtype of GBM that will consider and address the aforementioned shortcomings.

## Supporting information

**S1 Table. Detailed patient characteristics and datasets for treatment naïve glioma cohort.** (TIF)

**S2 Table. Detailed patient characteristics and datasets for recurrent GBM cohort.** (TIF)

**S3 Table. Antibody information and staining sequence.** (XLSX)

**S4 Table. Min and max number of cells per core in glioma and recurrent GBM TMAs.** (TIF)

**S1 Fig. Multiplexed immunofluorescence (MxIF) workflow.** The Cell DIVE MxIF workflow involves repeated cycles of staining, imaging and signal inactivation (panel A), following slide clearing and antigen retrieval. Prior to antibody staining, tissue is stained with DAPI and imaged in all channels of interest to record background autofluorescence (AF) of the tissue. Following background imaging, tissue is stained with 2–3 antibodies and reimaged to capture antigen-specific signal and then undergoes a dye inactivation step to remove the signal. The slide is re-imaged to measure background fluorescence intensity. These cycles are repeated multiple times until all targets of interest have been imaged. Panel 2 shows various image processing steps prior to generating single cell data. Some of these are performed during imaging itself while others are performed post image acquisition. The steps include, illumination correction, to correct for uneven illumination across the FOV, registration of images from all rounds (using DAPI signal from each round) and tissue AF removal. Panel C: Staining intensity of various cellular and subcellular markers is used to generate cellular segmentation masks. Segmented images are compared with real or virtual H&Es (generated from DAPI stained background images at the beginning of multiplexing) by a trained biologist or pathologist, and images with poor segmentation are removed from analysis. In parallel, marker staining is evaluated by reviewing AF removed images and markers that failed to stain or images with large artefacts are removed from analysis. Marker expression is quantified at cellular and subcellular compartments and data is generated in an easy to use .csv or Excel format which is then analyzed by a variety of different tools/approaches including simple statistical correlations, cluster analysis as well as heterogeneity analysis. (TIF)

**S2 Fig. Antibody validation workflow.** A typical antibody validation workflow: Starting with literature reports to identify antibody clones previously used for IHC on FFPE tissue, 3 or

more clones per target are identified and evaluated for sensitivity and specificity of the signal on a multi-tissue array (TMA) comprising all major tumor types and corresponding normal tissues. The down-selected antibody is conjugated with CY3, Cy5 or Cy7 at 2 different dye/ protein ratio and conjugates validated by staining comparison with unconjugated primary on serial sections of the same TMA. The down-selected conjugate is tested at different concentrations on a TMA with tumor tissue of interest to determine the optimal concentration for staining. In parallel, a set of TMA serial sections are pre-treated with different rounds of bleaching and evaluated for bleaching solution's effect on antigen of interest by comparing the staining among this set. Antigens with discernible effects are prioritized for staining early in the sequence, immediately after primary secondary staining of targets which failed to conjugate. (TIF)

**S3 Fig. Marker Staining quality assessment.** A: Marker staining performance in each cohort (True-positive, False-negative), staining round, subcellular location used for analysis and gene symbol, B: examples of quantitative FOV level correlation of marker intensities on replicate slides (Pearson correlation coefficients are shown), C: Examples of fluorescence image overlays of various hallmark markers showing heterogeneity of expression in astrocytoma. (TIFF)

**S4 Fig. Number of segmented cells in serial sections.** High correlation in number of segmented cells was observed between serial sections, particularly for the treatment naïve glioma cohort and two out of three sections of the recurrent GBM cohort. The Pearson correlation coefficient was computed and is presented in each slide to slide correlation plot. (TIF)

**S5 Fig. Example workflow for calculating cell molecular state and cell spatial heterogeneity.** Example of how molecular state and cell spatial heterogeneity metrics are calculated, using EGFR as an example. A. Segmentation of cells using DAPI staining and generation of nuclear and extra-nuclear masks; B. EGFR fluorescence intensity is quantified for each cell and discretized as low, moderate, and high. The different levels of cell expression are shown as red (high), green (moderate) or blue (low). C. For each cell (I through v in this cartoon), adjacent neighboring (touching) cells are counted, and their Spatial State is used to sum the Spatial Heterogeneity. (TIF)

**S6 Fig. Uni- (A) and multivariate (B) analysis of biomarker expression and overall survival as a function of IDH mutation status** A. Differences in individual biomarker expression and survival of IDHmt and IDHwt patients. B. A predictive multivariate model of IDH mutation status. (TIFF)

**S7 Fig. Lollipop plots for biomarker expression in each cluster, relative to population median.** Protein expression profiles of individual clusters plotted relative to median expression in the whole population. Solid circles represent the average expression in the cluster while direction and length of the lollipop shows difference in expression relative to population median (left-lower, right-higher). (TIF)

**S8 Fig. Cell clusters based on angiogenesis hallmark proteins.** Unsupervised clustering of cells using angiogenesis hallmark proteins identified a 5 cluster set. Clusters with lower than average hallmark protein expression (1 & 2) are highly represented in samples with IDH1 mutation. Cluster 4 & 5 with higher expression are proportionally more abundant in IDH1wt

samples.
(TIF)

**S9 Fig. Abundance of distinguisher genes (mRNA)/class per patient.** A: Relative proportion of cells belonging to different CellDistinguisher classes in each sample. Class 3 is highly represented in IDHwt samples. B: shows relative abundance of distinguisher genes grouped by hallmarks in individual classes.
(TIF)

**S10 Fig. Molecular and spatial heterogeneity in grade III gliomas and recurrent GBM IDHwt and IDHmt tumors.** Molecular and spatial heterogeneity in grade III gliomas and recurrent GBM IDHwt and IDHmt tumors according to the following hallmarks: Invasion and Motility, Cell Proliferative Signaling and Inducing Angiogenesis. The Mann-Whitney test was performed on all unpaired comparisons with the resulting p-value presented in each plot. The p-values were not adjusted for multiple testing.
(TIF)

**S11 Fig. Differences in MR features across the population range of RNA and protein marker expression for angiogenesis.** Correlation between Normalized enhancing core volume (derived from MRI) and Angiogenesis estimated from (a,d) RNA expression levels, and, based on multiplex immunofluorescence (MxIF) angiogenesis markers (b, e) S100A4 and (c,f) VEGFR2; (a-c) shows the plots on Cohort 1 (CW Glioma, treatment naive) while (d-f) show cohort 2 (UCSF, recurrent GBM). A progressive increasing trend may be observed in both cohorts when examining the normalized enhancing core volume for low, medium and high angiogenesis. The trends across the enhancement ratio are also conserved when comparing RNA with MxIF Angiogenesis. None of these comparisons reach statistical significance after multiple comparison correction using false discovery rate.
(TIF)

## Acknowledgments

George Reid assembled the tissue microarrays of primary glioma cases and reference tissues.

## Author Contributions

**Conceptualization:** Michael E. Berens, Anup Sood, Jill S. Barnholtz-Sloan.

**Data curation:** Michael E. Berens, Anup Sood, Jill S. Barnholtz-Sloan, John F. Graf, Sanghee Cho, Seungchan Kim, Jeffrey Kiefer, Sara A. Byron, Rebecca F. Halperin, Sara Nasser, Jonathan Adkins, Lori Cuyugan, Karen Devine, Quinn Ostrom, Marta Couce, Leo Wolansky, Elizabeth McDonough, Shannon Schyberg, Sean Dinn, Yousef Al-Kofahi.

**Formal analysis:** Michael E. Berens, Anup Sood, Jill S. Barnholtz-Sloan, John F. Graf, Sanghee Cho, Seungchan Kim, Jeffrey Kiefer, Sara A. Byron, Rebecca F. Halperin, Sara Nasser, Karen Devine, Quinn Ostrom, Marta Couce, Leo Wolansky.

**Investigation:** Michael E. Berens, Anup Sood, Jill S. Barnholtz-Sloan, John F. Graf, Andrew E. Sloan, Michael Prados, Joanna J. Phillips, Mirabela Rusu, Maria I. Zavodszky, Fiona Ginty.

**Supervision:** Michael E. Berens, Anup Sood, Jill S. Barnholtz-Sloan.

**Validation:** Elizabeth McDonough, Shannon Schyberg, Sean Dinn.

**Writing – original draft:** Michael E. Berens, Anup Sood, Jill S. Barnholtz-Sloan, Sarah J. Nelson, Winnie S. Liang, Yousef Al-Kofahi.

**Writing – review & editing:** Michael E. Berens, Anup Sood, Jill S. Barnholtz-Sloan, John F. Graf, Sanghee Cho, Seungchan Kim, Jeffrey Kiefer, Sara A. Byron, Rebecca F. Halperin, Sara Nasser, Karen Devine, Quinn Ostrom, Marta Couce, Leo Wolansky, Andrew E. Sloan, Michael Prados, Joanna J. Phillips, Sarah J. Nelson, Winnie S. Liang, Mirabela Rusu, Maria I. Zavodszky, Fiona Ginty.

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
