## [Decision Letter · Decision Letter 0]

20 Aug 2019

Bélatelep, Hungary

August 19, 2019

PONE-D-19-18177

Tumor cell phenotype and heterogeneity differences in IDH1 mutant vs wild-type gliomas

PLOS ONE

Dear Dr. Berens,

Thank you for submitting your manuscript to PLOS ONE. After careful consideration, we feel that it has merit but does not fully meet PLOS ONE’s publication criteria as it currently stands. Therefore, we invite you to submit a revised version of the manuscript that addresses the points raised by the Reviewer, listed below.

We would appreciate receiving your revised manuscript by Oct 04 2019 11:59PM. To enhance the reproducibility of your results, we recommend that if applicable you deposit your laboratory protocols in protocols.io, where a protocol can be assigned its own identifier (DOI) such that it can be cited independently in the future. For instructions see: http://journals.plos.org/plosone/s/submission-guidelines#loc-laboratory-protocols

We look forward to receiving your revised manuscript.

Kind regards,

Joseph Najbauer, Ph.D.

Academic Editor

PLOS ONE

Journal Requirements:

1. Please include your tables as part of your main manuscript and remove the individual files. Please note that supplementary tables (should remain/ be uploaded) as separate "supporting information" files

Reviewers' comments:

Reviewer's Responses to Questions

**Comments to the Author**

1. Is the manuscript technically sound, and do the data support the conclusions?

Reviewer #1: Yes

2. Has the statistical analysis been performed appropriately and rigorously? 

Reviewer #1: No

3. Have the authors made all data underlying the findings in their manuscript fully available?

Reviewer #1: No

4. Is the manuscript presented in an intelligible fashion and written in standard English?

Reviewer #1: Yes

5. Review Comments to the Author

Reviewer #1: Based on my review, these results have not been published elsewhere (except Biorxiv preprint).

In this study, the authors have used multiple platforms to investigate phenotype and heterogeneity differences in IDH1 mutant vs wild-type gliomas. Some interesting findings are presented and the study represents an essential contribution to the field. The approach for developing multiscale and multi parametric data to study differences in glioma subtypes and the way to assemble the analysis is the convincing and novel part of the study.

The manuscript contains large number of figures and tables demanding a more thorough presentation and conformity. This will make it easier to keep a track on the story. There are some gaps in the development of the arguments and some of the details/discussion/explanations were difficult to find.

Following are the points to work on and need to be answered.

1) Experimental statistics are not performed to a technical standard and are not described in sufficient detail. There is not much information about the statistical analysis in the whole manuscript and the figures legends do not have any details about the statistics performed such as tests, p value (not everywhere), significance and the correction (if used). (for example figure 1, 6 and S6A and S11). The author should provide a section in materials and method for detailed statistical analysis and software used.

2) The authors should provide the gene/protein name of the lollipop bars in Figure 3B and 3E (where the other genes are unchanged or slightly changed) for transparency.

3) In the Figure 5A and B, where author is comparing the cell dive biomarker intensity with bulk mRNA expression, there is high contradiction in the FASN and GSK3b expression. The author should also explain or discuss about the difference in expression values coming from cell dive and mRNA seq data for FASN and GSK3B. Western blot should be performed to verify the difference in the expression level of FASN and GSK3B in IDHwt and IDHmt tumors.

4) Figure 3A(h) where FASN overlaid on DAPI, DAPI staining does not look convincing (surprisingly, not all the cells were positive for DAPI in 3A(h) while in figure 3D (h) the DAPI staining looks fine). Any explanation?

5) (in the provided excel sheet) It is mentioned that false staining means staining failed or not observed in any of the cores and this were excluded from the analysis. There is a possibility that particular proteins are expressed in that tissue type. Could this be a source of bias? Did authors use positive controls tissue sections in these situations and were lack of staining with antibodies and presence of protein validated?

6) The mRNA seq data analysis could give many other interesting findings. The authors should also show the expression of few genes for CD8, NK cells, GD T cells, microglia/macrophages and MDSC cells in IDHmt and IDHwt, as they are important shares of the brain tumor microenvironment and contribute up to 50% of the total population.

7) The title of the paper should be changed and should mention the use of multiscale workflow/technology in the study. Especially, because major part of this study is describing the method and new way of analysis.

8) The abstract could be more informative on the specific findings of the study.

9) On the line 689 it says weather, it should be whether.

6. PLOS authors have the option to publish the peer review history of their article (what does this mean?). If published, this will include your full peer review and any attached files.

Reviewer #1: No

---

## [Author Response · Author response to Decision Letter 0]

23 Oct 2019

The critiques of the prior submission, as well as the authors’ responses/edits are noted below. 

1) Experimental statistics are not performed to a technical standard and are not described in sufficient detail. There is not much information about the statistical analysis in the whole manuscript and the figures legends do not have any details about the statistics performed such as tests, p value (not everywhere), significance and the correction (if used). (for example figure 1, 6 and S6A and S11). The author should provide a section in materials and method for detailed statistical analysis and software used.

>>> Yes, this was an oversight on our part. A new section on statistical analysis methods has been added at the end of the methods section. Here we have included the statistical methods and cited R software for biomarker, cluster, MR feature and heterogeneity comparison between IDHmt vs wt patients and survival analysis. Extensive detail was already provided on the k-means clustering and spatial analysis methods. 

2) The authors should provide the gene/protein name of the lollipop bars in Figure 3B and 3E (where the other genes are unchanged or slightly changed) for transparency.

 >> >Figure 3 has now been updated with all the protein names 

3) In the Figure 5A and B, where author is comparing the cell dive biomarker intensity with bulk mRNA expression, there is high contradiction in the FASN and GSK3b expression. The author should also explain or discuss about the difference in expression values coming from cell dive and mRNA seq data for FASN and GSK3B. Western blot should be performed to verify the difference in the expression level of FASN and GSK3B in IDHwt and IDHmt tumors.

>>> Despite the differences for FASN, GSK3b and NCad, the concordance between the two technologies is notable for most other markers, particularly given the differences in sample amount and preservation (fixed vs frozen). Otherwise, directional discordance between mRNA and protein for any given gene is not an unanticipated finding, and on an individual gene level, is more discordant in tumor versus cognate normal tissue (Idit Kosti, Nishant Jain, Dvir Aran, Atul J. Butte & Marina Sirota. Cross-tissue Analysis of Gene and Protein Expression in Normal and Cancer Tissues. 

Scientific Reports volume 6, Article number: 24799 (2016). Underlying reasons for this discordance include variant turnover rate of the protein versus mRNA, translational efficiency due to ribosomal density and occupancy, RNA secondary structure, among others (Correlation of mRNA and Protein in complex biological samples. 2009). Unfortunately it is not possible to conduct additional western analysis due to sample depletion. These points have also now been noted in the revised paper. 

4) Figure 3A(h) where FASN overlaid on DAPI, DAPI staining does not look convincing (surprisingly, not all the cells were positive for DAPI in 3A(h) while in figure 3D (h) the DAPI staining looks fine). Any explanation? 

>>>Since all images in the same group are set to same window/level in ImageJ, DAPI staining in 3A(h) is not readily apparent, however, the relative DAPI intensities between two images are different. Since these are two different patients the differences may be due to fixation conditions or age of the sample. 

5) (in the provided excel sheet) It is mentioned that false staining means staining failed or not observed in any of the cores and this were excluded from the analysis. There is a possibility that particular proteins are expressed in that tissue type. Could this be a source of bias? Did authors use positive controls tissue sections in these situations and were lack of staining with antibodies and presence of protein validated?

>>>Yes, a control TMA containing both positive and negative controls used for antibody validations were included in the multiplexing study to confirm antibody performance (described in the methods section, lines 197 and 201). Additionally, “Control cores were included on all slides for glioma, prostate, melanoma, lung, breast cancer (two per cancer type) to verify antibody performance.” (line 183-184). Her2 was not expected to stain positive, nor cytokeratin and many of the other markers with negative staining were immune related, suggesting perhaps an immunosuppressive environment. 

6) The mRNA seq data analysis could give many other interesting findings. The authors should also show the expression of few genes for CD8, NK cells, GD T cells, microglia/macrophages and MDSC cells in IDHmt and IDHwt, as they are important shares of the brain tumor microenvironment and contribute up to 50% of the total population.

>>> We agree the RNA Seq data could provide other findings. Our focus in this paper was to investigate the RNA and protein expression profiles and metrics at a pathway/hallmark level and compare those with MR features. However, immune evasion hallmark was not found to be a significantly correlated with MR features for either this cohort, nor in an earlier TCGA analysis of RNA and MR data from the Cancer Imaging Archive. Data will be made available for further analysis. 

7) The title of the paper should be changed and should mention the use of multiscale workflow/technology in the study. Especially, because major part of this study is describing the method and new way of analysis.

>>> Appreciate the helpful suggestion, we have changed the title to: Multiscale, multimodal analysis of tumor heterogeneity in IDH1 mutant vs wild-type diffuse gliomas

8) The abstract could be more informative on the specific findings of the study.

>>> The information content of the abstract now contains additional specific findings of the study

9) On the line 689 it says weather, it should be whether. 

>> Changed

---

## [Decision Letter · Decision Letter 1]

13 Nov 2019

Pécs, Hungary

November 12, 2019

Multiscale, multimodal analysis of tumor heterogeneity in IDH1 mutant vs wild-type diffuse gliomas

PONE-D-19-18177R1

Dear Dr. Berens,

We are pleased to inform you that your manuscript (R1 version) has been judged scientifically suitable for publication and will be formally accepted for publication once it complies with all outstanding technical requirements.

With kind regards,

Joseph Najbauer, Ph.D.

Academic Editor

PLOS ONE

Reviewers' comments:

Reviewer's Responses to Questions

**Comments to the Author**

1. If the authors have adequately addressed your comments raised in a previous round of review and you feel that this manuscript is now acceptable for publication, you may indicate that here to bypass the “Comments to the Author” section, enter your conflict of interest statement in the “Confidential to Editor” section, and submit your "Accept" recommendation.

Reviewer #1: All comments have been addressed

2. Is the manuscript technically sound, and do the data support the conclusions?

Reviewer #1: Yes

3. Has the statistical analysis been performed appropriately and rigorously? 

Reviewer #1: Yes

4. Have the authors made all data underlying the findings in their manuscript fully available?

Reviewer #1: Yes

5. Is the manuscript presented in an intelligible fashion and written in standard English?

Reviewer #1: Yes

6. Review Comments to the Author

Reviewer #1: (No Response)

7. PLOS authors have the option to publish the peer review history of their article (what does this mean?). If published, this will include your full peer review and any attached files.

Reviewer #1: No

---

## [Editor Report · Acceptance letter]

12 Dec 2019

PONE-D-19-18177R1 

Multiscale, multimodal analysis of tumor heterogeneity in IDH1 mutant vs wild-type diffuse gliomas 

Dear Dr. Berens:

I am pleased to inform you that your manuscript has been deemed suitable for publication in PLOS ONE. Congratulations! Your manuscript is now with our production department. 

With kind regards,

on behalf of

Dr. Joseph Najbauer 

Academic Editor

PLOS ONE